# TRAIP resolves DNA replication-transcription conflicts during the S-phase of unperturbed cells

Shaun Scaramuzza[1,4], Rebecca M. Jones [1,6], Martina Muste Sadurni[1,6], Alicja Reynolds-Winczura[1], Divyasree Poovathumkadavil[1], Abigail Farrell [1], Toyoaki Natsume [2,3,5], Patricia Rojas[1], Cyntia Fernandez Cuesta[1], Masato T. Kanemaki [2,3], Marco Saponaro[1] & Agnieszka Gambus [1] ✉

Cell division is the basis for the propagation of life and requires accurate duplication of all genetic information. DNA damage created during replication (replication stress) is a major cause of cancer, premature aging and a spectrum of other human disorders. Over the years, TRAIP E3 ubiquitin ligase has been shown to play a role in various cellular processes that govern genome integrity and faultless segregation. TRAIP is essential for cell viability, and mutations in TRAIP ubiquitin ligase activity lead to primordial dwarfism in patients. Here, we have determined the mechanism of inhibition of cell proliferation in TRAIP-depleted cells. We have taken advantage of the auxin induced degron system to rapidly degrade TRAIP within cells and to dissect the importance of various functions of TRAIP in different stages of the cell cycle. We conclude that upon rapid TRAIP degradation, specifically in S-phase, cells cease to proliferate, arrest in G2 stage of the cell cycle and undergo senescence. Our findings reveal that TRAIP works in S-phase to prevent DNA damage at transcription start sites, caused by replication-transcription conflicts.

Cell proliferation is the basis for the propagation of life and requires accurate duplication of all genetic information before correct cell division. However, these processes encounter impediments that threaten their faultless execution and lead to genomic instability. Moreover, problems encountered during DNA replication (replication stress) often result in under-replicated DNA that needs to be resolved during mitosis. Cells have developed a number of means to respond to these challenges, such as replication-coupled DNA repair pathways and S-phase checkpoint responses that protect the stability of replication forks, inhibit the initiation of replication in new areas of the genome, and block cell cycle progression to allow for DNA damage resolution[1]. Insufficiencies in these S-phase responses and persistence of unreplicated DNA past S-phase, induce rescue mechanisms during mitosis, such as mitotic DNA synthesis (MiDAS) and anaphase-bridge resolution. Failure of all these pathways results in DNA breakage, chromosome missegregation, and chromosomal rearrangements. Genome-wide, such unreplicated regions correlate with common fragile sites (CFS), responsible for recurrent re-arrangements often found in human disease[2,3].

TRAIP (TRAF-interacting protein, also known as TRIP or RNF206) E3 ubiquitin ligase has recently been shown to play a role in a number of the above processes. TRAIP is essential for cell proliferation at an early stage of development in the mouse embryo[4] and CRISPR/Cas9-mediated deletion of TRAIP in a number of human cell

[1]Institute of Cancer and Genomic Sciences, Birmingham Centre for Genome Biology, University of Birmingham, Birmingham, UK. [2]Department of Chromosome Science, National Institute of Genetics, Research Organization of Information and Systems, Mishima, Shizuoka, Japan. [3]Department of Genetics, The Graduate University for Advanced Studies (SOKENDAI), Mishima, Shizuoka, Japan. [4]Present address: Cancer Research UK – Manchester Institute, Manchester Cancer Research Centre, Manchester, UK. [5]Present address: Research Center for Genome & Medical Sciences, Tokyo Metropolitan Institute of Medical Science, Tokyo, Japan. [6]These authors contributed equally: Rebecca M. Jones, Martina Muste Sadurni. ✉e-mail: a.gambus@bham.ac.uk

lines is lethal[5]. Moreover, homozygous mutations of the TRAIP ubiquitin ligase domain in humans lead to microcephalic primordial dwarfism[6].

At the cellular level, TRAIP has been shown to be essential for the appropriate repair of DNA damage caused by mitomycin C (MMC) and other inter-strand crosslinks (ICLs) generating drugs: camptothecin (CPT), UV, and hydroxyurea (HU)[6-9]. TRAIP has also been reported to regulate mitotic progression; cells with downregulated TRAIP go through mitosis faster and with more chromosome segregation errors[10,11]. TRAIP usually accumulates in nucleoli in cells and re-localises to sites of DNA damage and replication stress[6-8,12]. This co-localisation with PCNA upon DNA damage is mediated by a PCNA-interacting protein (PIP)-box motif located at the C-terminus of TRAIP[7,8]. However, TRAIP was also shown to interact with unchallenged replication forks in *Xenopus* egg extract[13] and, in human cells, TRAIP has been shown to interact with nascent DNA during unperturbed S-phase through Nascent Chromatin Capture (NCC) and iPOND[7,14,15].

At the molecular level, TRAIP has been shown to orchestrate the response to ICLs in *Xenopus laevis* egg extract through the ubiquitylation of the eukaryotic replicative helicase (CMG complex, from CDC45/MCM2-7/GINS) as two replication forks converge at the ICL[13]. In such a situation, short ubiquitin chains synthesised by TRAIP on CMGs promote the recruitment of NEIL3 glycosylase and unhooking of the ICL, whilst longer ubiquitin chains are required for CMG unloading by p97 segregase, allowing access for endonucleases and Fanconi anaemia pathway proteins that perform ICL repair[13,16,17]. Using the same model system, TRAIP has also been shown to act when the replisome encounters DNA-protein crosslinks (DPCs), which impair replication forks progression. In such a situation, however, TRAIP ubiquitylates not the CMG helicase within the blocked replisome, but the protein barrier itself[18]. Moreover, in human cells, TRAIP was also shown to interact with RNF20-RNF40 ubiquitin ligase at double-strand breaks and affect ionizing-radiation induced monoubiquitylation of histone H2B[19,20], as well as the localisation of BRCA1 interacting partner RAP80 to DNA double-strand breaks[19].

On the other hand, in mitosis, in *Xenopus leavis* egg extract, *C. elegans* embryos, and mouse embryonic stem cells, we and others have shown that TRAIP ubiquitylates any replicative helicases left on chromatin from replication in S-phase, leading to their unloading by the p97 segregase[21-24]. Such replisomes retained on chromatin until mitosis likely protect the DNA they are bound to, preventing access and subsequent DNA processing by nucleases. As a result, TRAIP-driven replisome disassembly in mitosis can lead to fork breakage and complex DNA re-arrangements in *Xenopus* egg extract[23] and allows for MiDAS in human cells[22]. Finally, TRAIP depletion was reported to lead to decreased stability of kinetochore-microtubule attachments and diminished spindle assembly checkpoint function through lowered MAD2 levels at centromeres[10,25].

With so many varied functions of TRAIP reported, the question arises: which one of its functions is most crucial for cell viability, proliferation, and prevention of microcephalic dwarfism? To answer this, we have generated auxin-inducible degrons of TRAIP in the colon carcinoma HCT116 cell lines and immortalised retinal pigment epithelial cells hTERT-RPE1, which facilitate the degradation of TRAIP within 30 min of auxin (IAA) addition to the cell media. With these, we could confirm that upon rapid TRAIP degradation, cells cease to proliferate, arrest in G2 stage of the cell cycle and undergo senescence. By further investigating the effect of TRAIP degradation on specific stages of the cell cycle we found that TRAIP plays an essential role in S-phase and that the lack of TRAIP results in the generation of DNA damage at sites of replication-transcription collisions. We propose a model whereby TRAIP acts at sites of collision and allows the conflicts to be resolved.

## Results

To investigate the role of TRAIP during the unperturbed eukaryotic cell cycle, we established conditional auxin-inducible degron (AID) cell lines[26,27]. As modifying the N-terminus of TRAIP was reported to impact TRAIP localisation in the cell[8], endogenous TRAIP was tagged C-terminally in an HCT116 cell line expressing OsTIR1, an E3 ligase component recognising the degron, with either a mini-auxin-inducible degron (mAID) tag, or an mCLOVER-fused variant (mAC); henceforth referred to as TRAIP-mAID and TRAIP-mAC, respectively (Supplementary Fig. 1A, B). Two clones of each HCT116 TRAIP-mAID and TRAIP-mAC were generated and used throughout this work to ensure that the observed phenotypes are not CRISPR off-target effects. The hTERT-RPE1 cell line chosen for TRAIP modification expressed OsTIR1 from an inducible Tet promoter, requiring the addition of tetracycline or doxycycline to express it (Supplementary Fig. 1A). Bi-allelic tagging was verified by PCR amplification from the genomic *TRAIP* locus (Supplementary Fig. 1C, D). It was also confirmed at the protein level by the disappearance of the endogenous untagged TRAIP protein from the whole cell extracts prepared using degron cell lines (Supplementary Fig. 1E). Bi-allelic modification of *TRAIP* with degron tags did not affect cell proliferation and fitness (Supplementary Fig. 1F, G). Activation of the AID system, through the addition of Indole-3-acetic acid (auxin, IAA) to the cell growth media ensured rapid protein degradation in ~30 minutes (Fig. 1A).

### TRAIP activity is essential for cell proliferation

We first set out to understand the global consequences of TRAIP depletion for cell viability, proliferation, and cell cycle progression. Colony assays were used to assess the respective impact of auxin treatment itself, and TRAIP degradation, on cell viability. Long-term auxin treatment (7–14 days) led to abrogation of colony growth in all the conditional TRAIP degron cell lines (Fig. 1B; Supplementary Fig. 2A). No reductions in the overall colony forming ability were found when auxin was added to control cell lines, indicating that the drug treatment itself was not cytotoxic. These data confirmed that TRAIP was an essential protein as previously described[4,5]. Importantly, we could significantly rescue the cell growth in the auxin-treated HCT116 TRAIP-mAID degron cells by expression of exogenous full-length (wild-type, wt) TRAIP, but not an active site mutant C7A/C10A of TRAIP or TRAIP W37A[6] (Fig. 1C). This indicates that TRAIP ubiquitin ligase activity is essential for cell viability in our system. TRAIP expression is highly regulated in the cell and overexpression of wtTRAIP was reported to be toxic[7], which is a likely reason why the expression of wtTRAIP in our experiments does not fully rescue lethality of the TRAIP-mAID degradation.

To investigate the underlying reason for the observed growth inhibition, we next investigated the effects of shorter periods of protein degradation (24-72 Hrs) on cell proliferation and cell cycle progression. HCT116 cells lacking TRAIP exhibited clear defects to cell growth at all time points tested (Fig. 1D, Supplementary Fig. 2B), while RPE1 cells showed less dramatic defect (Supplementary Fig. 2C). Again, we could rescue the proliferation defect observed after 72 h of TRAIP-mAID degradation in HCT116 cells by expression of wtTRAIP, but not TRAIP C7A/C10A or TRAIP W37A (Fig. 1E). Furthermore, flow cytometric analysis revealed a reduction in the number of cells in G1 stage of the cell cycle, as well as the accumulation of TRAIP-depleted cells with G2/M DNA content (Fig. 1F, Supplementary Fig. 2D, E), which could be rescued in HCT116 cells by expression of wtTRAIP but not the ubiquitin ligase dead mutants (Fig. 1G). Altogether, these results show that our TRAIP degron cell lines exhibit phenotypes analogous to those observed previously upon downregulation of TRAIP by siRNA or in patients' cells with impaired ubiquitin ligase activity of TRAIP[6-8] and thus provide a useful tool for the determination of the cell-cycle, stage-specific and essential function of TRAIP. Moreover, the essential function of TRAIP depends on its ubiquitin ligase activity.

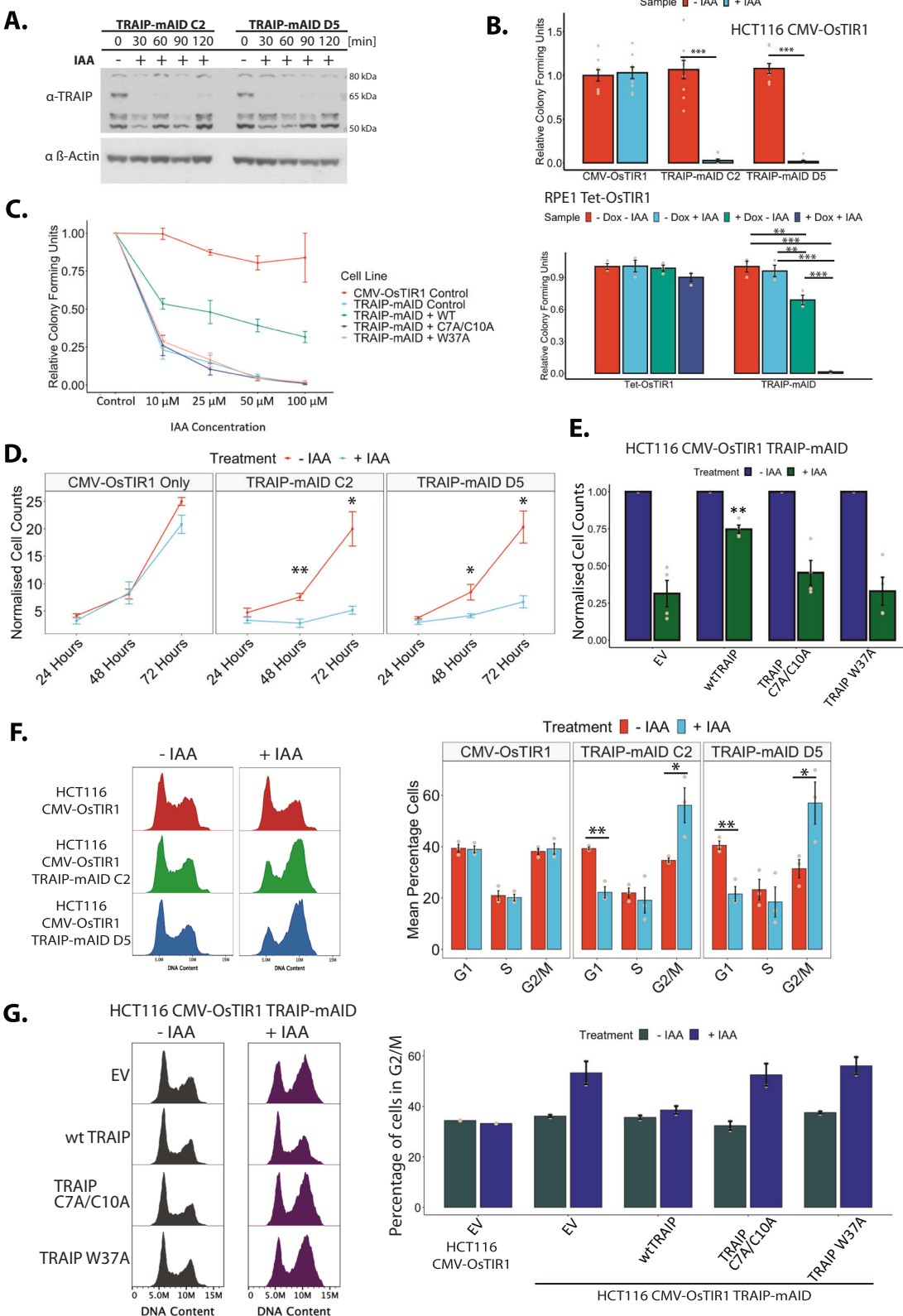

## TRAIP-depleted cells accumulate in G2 stage of the cell cycle

As the underlying mechanism of cell death in cells depleted of TRAIP has not been previously determined, we decided first to investigate the mechanism by which TRAIP-depleted cells stop proliferating. We first determined in which cell cycle stage cells lacking TRAIP were accumulating: G2 or mitosis. We focused on HCT116 cells, as they proliferate faster and at higher cell densities, facilitating cell cycle analysis by flow cytometry and easier detection of cell cycle defects. Quantification of the total proportion of cells displaying phosphorylation of Histone H3 on S10 (marker of mitosis) by flow cytometry showed no differences to the numbers of mitotic cells following TRAIP degradation (24 h auxin treatment) (Supplementary Fig. 3A). The same was confirmed by mitotic indexing experiments (Supplementary Fig. 3B, no WEE1i). To specifically visualise the G2 population of cells, cells

**Fig. 1 | Rapid auxin-induced degradation of TRAIP-mAID leads to G2 cell cycle arrest and inhibition of cell proliferation. A** An asynchronous cell culture of HCT116 TRAIP-mAID cells was supplemented with auxin (IAA) and samples taken at indicated times. The level of TRAIP-mAID was assessed by western blotting with TRAIP antibody. Two independent clones of TRAIP-mAID are presented ($n = 3$). **B** Cells were grown in presence or absence of auxin (IAA) and in case of hTERT-RPE1 cells in presence or absence of Doxycycline. The resultant colonies were visualised. Quantification of three biological repeats; depicted as the mean normalised colonies formed ±SEM. Pairwise hypothesis testing was conducted using Mann–Whitney $U$ tests, with significance values indicated on the plot (HCT116 Cells: TRAIP-mAID C2: $p < 0.001$; TRAIP-mAID D5: $p < 0.001$; RPE1 Cells: significant differences were observed between TRAIP-mAID cell lines only. -Dox −IAA vs +Dox −IAA: $p = 0.004$; −Dox −IAA vs +Dox +IAA: $p = <0.001$) **C** HCT116 TRAIP-mAID cells were transfected with retroviruses expressing WT, C7A/C10A or W37A mutants of TRAIP. Colony assay viability was assessed at different concentrations of auxin. Quantification of $n = 3$. Error bars depicted show the mean colony forming units ±SEM. **D** TRAIP-mAID cells were grown in optional presence of auxin (IAA) and cell proliferation counted over the 72 h. Quantification of $n = 3$ for two independent TRAIP-mAID clones. Data is shown as the mean ±SEM. Pairwise statistical comparison were conducted at each timepoint using t.tests, with significance indicated on the plot (TRAIP-mAID C2: 48 h − $p = 0.00940$, 72 h − $p = 0.0363$; TRAIP-mAID D5: 48 h − $p = 0.04$, 72 h − $p = 0.0305$). **E** HCT116 TRAIP-mAID cells expressing different versions of TRAIP as in **C** were grown in presence of auxin for 72 h and counted. Data shown as the mean cell counts (normalised to control) ±SEM. Statistical significance was conducted to compare each +IAA treatment between cell lines (Empty Vector +IAA vs WT TRAIP + IAA, $p = 0.01211$. **F** TRAIP-mAID cells (two independent clones) were grown for 24 h in optional presence of auxin (IAA) and analysed for their DNA content by FACS. Example FACS plots (left) and quantification of number of cells in different stages of the cell cycle (mean ±SEM) over three independent experiments for two independent clones (right) are shown. Pairwise comparisons were carried out using t.tests, significance is indicated on the plot (TRAIP-mAID C2: G1 − $p = 0.0120$, G2/M − $p = 0.04$; TRAIP-mAID D5: G1 − $p = 0.00874$, G2/M − $p = 0.04$). **G** HCT116 TRAIP-mAID cells expressing different versions of TRAIP as in **C** were grown in presence of auxin for 24 h and analysed for their DNA content by flow cytometry. Example FACS plots (left) and quantification of number of cells in different stages of the cell cycle (mean ±SEM) over two independent experiments (right). Source data are provided as a Source Data file.

were treated with auxin for 24 hours to degrade endogenous TRAIP and stained with three markers of G2/M progression: DAPI for condensed chromosomes, Mitosin (CENPF) for G2 and mitosis, and the S10 phosphorylation of Histone H3 for mitosis (Fig. 2A). A significant increase was found in the proportions of cells in G2 stage of the cell cycle following treatment with auxin and TRAIP degradation (Fig. 2A, no WEE1i), while no differences again were found when considering those cells in either early or late mitosis (Supplementary Fig. 3C). Finally, the G2 cell cycle accumulation was overcome through inhibition of the G2 checkpoint kinase WEE1 using MK-1775 inhibitor (Fig. 2A and Supplementary Fig. 3C, D). We conclude, therefore, that the loss of TRAIP results in a G2 cell cycle arrest dependent on WEE1.

### G2-arrested TRAIP-depleted cells exit the cell cycle through senescence

Analyses of the HCT116 TRAIP-mAID cell line treated for 24 h with auxin also revealed an increase in the size of nuclei and overall cell size (Supplementary Fig. 4A, B). Intriguingly, increased cell size, reductions in cell proliferation, and cell cycle accumulation are all hallmarks of cell cycle exit via senescence[28]. To explore this further, we used flow cytometry to detect the senescence marker β-galactosidase. Indeed, following as little as 24 h of auxin treatment, cells were positive for this marker of cell senescence (Fig. 2B). Subsequent comparison of the positive β-galactosidase cell population with total DNA content revealed the specific activation of senescence in the accumulating G2/M cells (Fig. 2C). In contrast, we have detected no increase in the proportion of apoptotic cells (Supplementary Fig. 4C, D) and we observed no increase in cells exhibiting mitotic catastrophe phenotypes (data not shown).

### G2 arrest in TRAIP-depleted cells is a result of DNA damage checkpoint activation

G2 cell cycle arrest is often a consequence of DNA damage checkpoint activation[29] and TRAIP was previously linked to numerous DNA damage repair pathways. To check whether DNA damage accumulation upon degradation of TRAIP may be indeed responsible for the observed G2 arrest, cells were treated with auxin for 24 hours and stained for the general DNA damage marker γ-H2AX (S139 phosphorylation on H2AX) as well as the double-stranded DNA break-specific marker 53BP1. This revealed a general, but subtle, accumulation of DNA damage foci in the two independent HCT116 TRAIP-mAID cell lines (Fig. 2D) and in the hTERT-RPE1 TRAIP-mAID cells (Supplementary Fig. 5A). All these indicate that TRAIP is important for the maintenance of genomic stability. Moreover, inhibition of ATR, but not ATM, was able to rescue the observed G2 cell cycle arrest

(Supplementary Fig. 5B) suggesting that the damage is likely created during S or G2 stages of the cell cycle. We conclude, therefore, that the loss of TRAIP results in the creation of DNA damage, checkpoint activation, and the accumulation of cells in G2 stage of the cell cycle. As a result of this, cells enter senescence and stop proliferating.

### TRAIP is essential for completion of S-phase

To understand specifically when during the cell cycle TRAIP was essential, we combined the rapid nature of the AID system with established cell synchronisation techniques in HCT116 cells. Cells depleted of TRAIP accumulate in G2 stage of the cell cycle due to ATR checkpoint activation. TRAIP is known also to play a role in S-phase, we, therefore, started our investigations assessing whether TRAIP was essential during S-phase. Cells were arrested in G1 with lovastatin, and auxin added for the last hour of G1 arrest. First, we determined that there was no defect in cells entering S-phase (Supplementary Fig. 6A). Second, we assessed the relative impact of TRAIP depletion during S-phase on the cell cycle using the previously described G2 cell cycle arrest as a readout. When cells progressed through S-phase and into G2 lacking TRAIP, G2 cell cycle accumulation could be observed (Fig. 3A). Third, to verify whether TRAIP is required specifically during the G2 stage of the cell cycle, TRAIP was optionally degraded 15 hours post release from G1 arrest, when cells were in late S-phase. When auxin was added during late S-phase, no G2 cell cycle accumulation could be detected (Fig. 3B). These data indicated that TRAIP was essential specifically for mechanisms existing during S-phase.

Finally, TRAIP was previously reported to have many functions also during mitosis[10,11,21–24]. To determine whether we could observe that TRAIP regulates mitotic progression in our cells, we arrested cells specifically in G2 stage of the cell cycle with RO-3306, degraded TRAIP, released cells to mitosis, and followed chromosomal condensation and segregation (Supplementary Fig. 6B). In this we could observe that, irrespectively of TRAIP degradation, cells condensed chromosomes at a similar rate (Supplementary Fig. 6C). On the other hand, we found fewer chromosome segregating cells at 60 min post release into mitosis when TRAIP was absent (Supplementary Fig. 6D). This suggests that TRAIP is indeed regulating the timing of mitotic progression independently of its S-phase role. Therefore, to determine whether any functions of TRAIP in mitosis could contribute to the increased DNA damage response seen in the next S-phase, TRAIP was degraded specifically before mitosis, and its effect on the next cell cycle arrest was explored. Cells were arrested in G2 and treated with auxin for 1 hour prior to release into mitosis. Approximately 2 hours following release, when cells were in G1 phase of the cell cycle, auxin was optionally washed off to allow protein re-expression before entry into the next

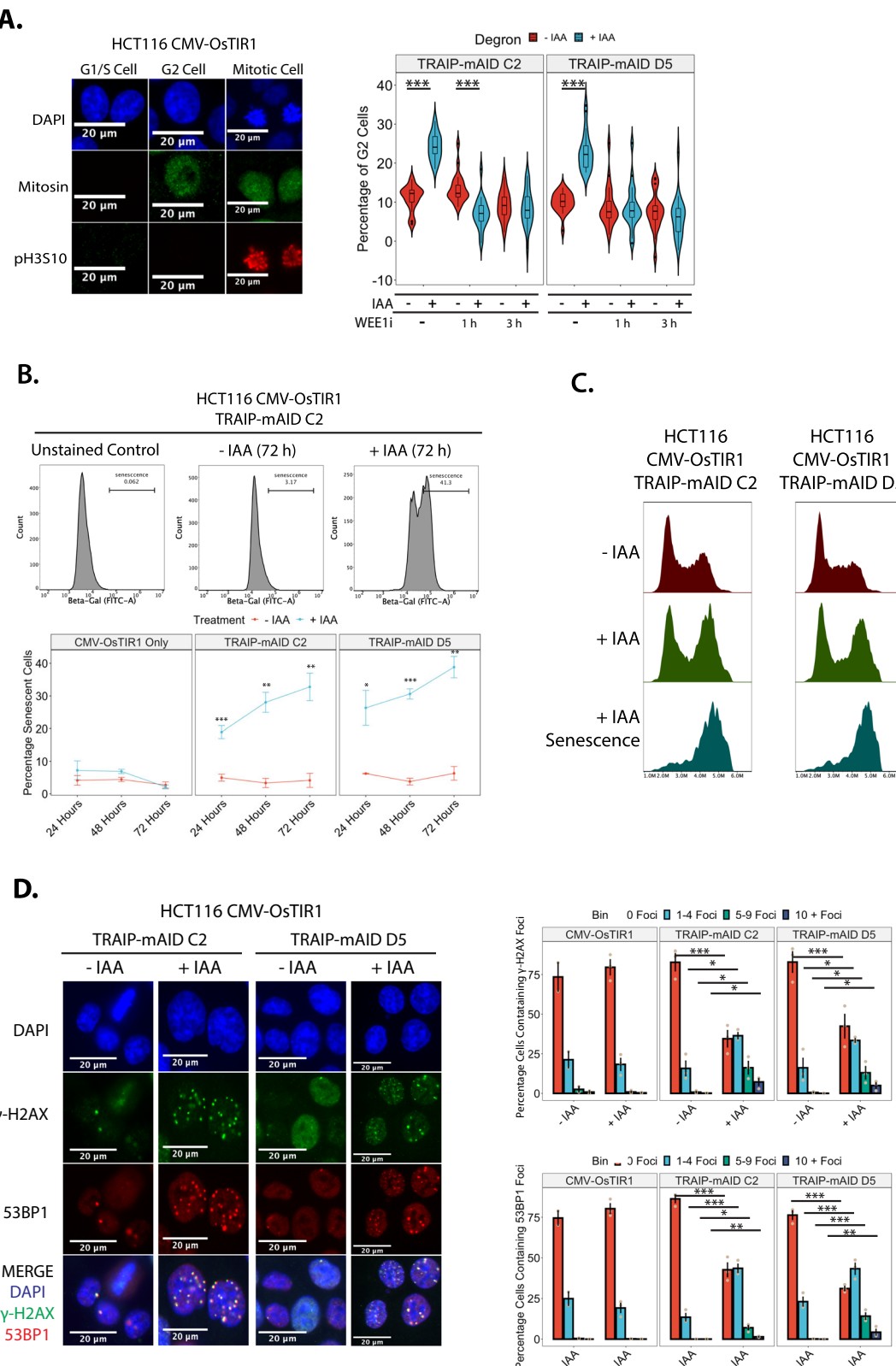

S-phase. Timepoints were taken 24 hours after release into mitosis. Using this experimental set-up, cell cycle accumulation in G2 was only seen when cells progressed through both mitosis and S-phase without TRAIP. No such accumulation was seen in those samples where TRAIP was allowed to re-express prior to S-phase entry (Fig. 3C). We, therefore, conclude that the essential function of TRAIP is executed during S-phase.

## TRAIP depletion leads to generation of DNA damage at transcription start sites (TSS)

Given the essential requirement of TRAIP during S-phase progression, we explored whether auxin treatment had any effect on global DNA replication. Cells treated with and without auxin for 24 hours were pulsed with the thymidine analogues EdU and BrdU for 1 hour, and analogue incorporation was analysed using immunofluorescence and

**Fig. 2 | TRAIP degradation leads to DNA damage, G2 cell cycle arrest and cell senescence. A** TRAIP-mAID cells (two independent clones) were grown for 24 h in optional presence of auxin (IAA), and optional addition of WEE1i for indicated times, and analysed for accumulation of cells in G2 (Mitosin staining) or in mitosis (pH3S10 staining). Quantification of three independent experiments. Data depicted as violin plots, displaying boxplots highlighting the median and corresponding interquartile ranges. Statistical analysis was carried out using one-way ANOVA and subsequent post hoc testing, significance is indicated on the plots (ANOVA: $p < 0.0001$. Tukey's post hoc tests: all indicated comparisons on the plot $p < 0.0001$). **B** TRAIP-mAID cells were grown for 24–72 h in optional presence of auxin (IAA) and the level of senescent cells analysed by FACS measurement of senescence marker β-galactosidase. Example FACS plots (top) and quantification of 3 independent experiments in 2 independent clones (bottom). Quantified data is shown as the mean ±SEM. Pairwise hypothesis testing was carried out using t.tests, with significance indicated on the plot (TRAIP-mAID C2: 24 h – $p = 0.00118$, 48 h –

$p = 0.00651$, 72 h – $p = 0.00926$; TRAIP-mAID D5: 24 h – $p = 0.03$, 48 h – $p = 0.00031$, 72 h – $p = 0.0022$). **C** TRAIP-mAID cells were grown for 72 h in optional presence of auxin (IAA). β-galactosidase staining was combined with DNA content analysis by FACS. β-galactosidase positive cells have G2/M DNA content. **D** TRAIP-mAID cells were grown in optional presence of auxin (IAA) for 24 h and stained with markers of DNA damage response (γH2AX and 53BP1). Example of immuno-fluorescent visualisation of DNA damage (left) and quantification of percentage of cells containing increasing numbers of γH2AX and 53BP1 foci over 3 independent experiments (right). Significance testing carried out using one-way ANOVA and subsequent Tukey's post hoc testing (TRAIP-mAID C2: ANOVA – $p < 0.001$; Tukey's post hoc test – 0 Foci: $p = <0.001$; 1–4 Foci: $p < 0.001$; 5-9 Foci: $p = 0.0168$; 10 + Foci: $p = 0.0059$. TRAIP-mAID D5: ANOVA – $p < 0.001$; Tukey's post hoc test – 0 Foci: $p < 0.001$; 1–4 Foci: $p < 0.001$; 5–9 Foci: $p = 0.0046$. Source data are provided as a Source Data file.

FACS respectively (Supplementary Fig. 7A, B). A lower number of cells was detected incorporating EdU/BrdU upon TRAIP degradation, suggesting fewer cells in S-phase. However, the cells that were in S-phase and labelled with BrdU or EdU, showed the same level of analogue incorporation, suggesting a similar rate of replication progression independent of TRAIP degradation status (Supplementary Fig. 7A, B). Based on our previous observations, it is likely that the lower proportions of cells in S-phase are due to the G2 cell cycle arrest and overall fewer cycling cells. TRAIP has also been previously implicated in replisome disassembly in model organisms during mitosis and stressed conditions[13,18,21–24]. We, therefore, wondered whether TRAIP was needed for global replisome disassembly during S-phase. The levels of chromatin-bound replicative helicase (MCM2-7) were assessed using flow cytometry following extraction of soluble proteins from permeabilised cells. No differences could be detected in either the loading or unloading of MCM2-7 during the otherwise unperturbed cell cycle (Supplementary Fig. 7C). Despite these results, the S-phase progression upon TRAIP degradation is not without problems. To investigate this further, cells progressing through S-phase with or without TRAIP were arrested in G2 with RO-3306 and pulsed with EdU to detect any late cell cycle DNA synthesis. When cells progressed through S-phase without TRAIP, our results suggest that they struggled to complete DNA synthesis on time as EdU incorporation could be detected in late G2/early mitosis (Supplementary Fig. 8A).

To gain insight into the mechanistic requirements for TRAIP during S-phase, we next used γ-H2AX ChIP-Seq to map genome-wide the location of the accumulating DNA damage. We first confirmed that the previously observed DNA damage accumulation was indeed specific to TRAIP's function during S-phase in both cell backgrounds (Supplementary Fig. 8B, C). HCT116 TRAIP-mAID cells were then prepared for sequencing at the 16-hour timepoint (late S-phase) as determined to display the maximal accumulation of DNA damage (Supplementary Fig. 8B). H2AX chromatin immuno-precipitation was used for sample normalisation, and we performed two independent repeats of this experiment. Reassuringly, the ChIP-Seq detected an overall increase to the γ-H2AX signal at a subset of genomic loci detected following auxin treatment in both repeats (example in Fig. 4A and Supplementary Fig. 9A). We first checked whether the position of γ-H2AX signal correlated with known DNA replication features: replication initiation sites and replication termination zones (both mapped in HCT116 cells by alternative techniques by[30] and[31]) and CFS[32]. However, no correlation could be found in either of the repeats (Fig. 4B and Supplementary Fig. 9B). Subsequently we performed a peak calling analysis to determine hotspots of DNA damage in the two repeats. We determined 1799 γ-H2AX signal peaks in the first experiment and 5984 in the second. 545 γ-H2AX signal peaks were common between both experiments (example of a hotspot common peak is presented in Fig. 4A). Interestingly, when we analysed the location of these 545 γ-H2AX hotspots, we found that they were largely associated with RNA Pol II transcripts. Indeed, we found that 95.77% of these peaks were

present on genes, and 88.24% present specifically around the TSS ±2 kb (Fig. 4C). We could also see a similar correlation with TSS for all γH2AX peaks called in both experiments, not only the common hotspots. Guided by this finding, we analysed the positions of all TRAIP degradation-induced γH2AX signals in both repeats in relation to all transcribed genes. Importantly, we found a general increase of γ-H2AX signal at TSS, even when the damage levels were not increased enough for stringent conserved peak calling analysis (Fig. 4D and Supplementary Fig. 9C). Hence, we focused on understanding what was special about the TSS of the genes exhibiting the hotspots of DNA damage upon TRAIP degradation. Firstly, we excluded an enrichment for genes involved in any particular cellular pathway through Go-term analysis (Supplementary Table 1). Next, we characterised replication dynamics across these sites using previously published replication data for HCT116[30], to determine the replication timing associated with these sites and we found that hotspot TSS sites are preferentially replicated in early S-phase (Supplementary Fig. 9D). We then identified the closest replication origin, as assigned by Koyanagi et al. to the γ-H2AX peak[31], determining also whether the replication fork will reach the TSS in a codirectional or head-to-head orientation with the RNA Pol II. We found that γ-H2AX peaks are close to their nearest origin, on average only 29 kb away (Fig. 4E). In comparison, TSS of genes that do not show an increase in γ-H2AX levels following TRAIP depletion (top 450 genes from Supplementary Fig 9C) are 41 kb away from their nearest origin (Fig. 4E). Moreover, γ-H2AX peaks occur preferentially around the TSS of the first transcribed gene encountered by the replication fork (75% of the total combining codirectional 1 and head-to-head 1), with the same frequency of replication fork and gene orientation being codirectional or head-to-head (Fig. 4F). In comparison, TSS of the genes with the lowest changes in γ-H2AX levels tend to occur less frequently on the first transcribed gene encountered by the replication fork (65% of the cases, Fig. 4F, chi-square test 0.034). We then overlayed this information with previously published datasets for precision run-on sequencing (PRO-Seq), tracking strand-specific nascent transcription activity[33], as well as RNA Pol II chromatin immuno-precipitation sequencing (ChIP-Seq)[34] in HCT116 cells. Interestingly, when we analysed the PRO-Seq signal at the hotspot TSS, we found that 98.25% of all the genes with a γ-H2AX hotspot presented clear levels of bi-directional transcription at their TSS, independently of the reciprocal orientation, either because of bi-directional promoters or TSS-associated antisense transcription. We measured therefore the extent of the antisense transcription by calculating the ratio between the levels of the antisense transcription and the sense transcription. To do this, we used the strand-specific PRO-Seq data, and determined for each gene the amount of antisense transcription occurring in the region –1000 bp -> TSS on the opposite strand of the gene. This was then divided by the amount of sense transcription TSS -> +1000 bp. We found that, compared to all the rest of the transcribed genes, those with a γ-H2AX hotspot peak presented higher levels of antisense transcription at their TSS (Fig. 4G). We also did the same analysis with

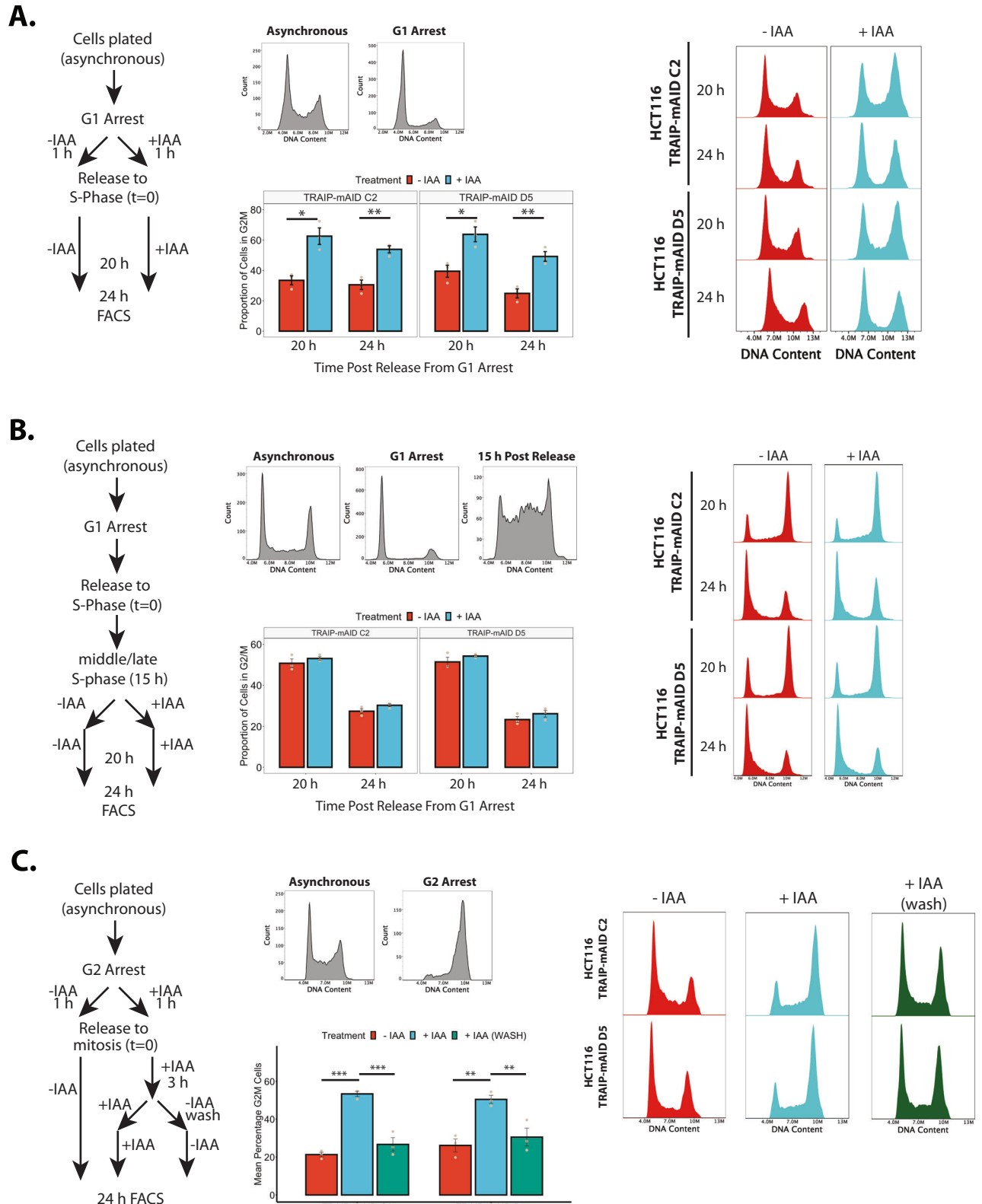

total RNA Pol II ChIP-Seq data[34] finding the same result (Fig. 4G, Supplementary Fig. 9E). In comparison, genes with the lowest changes of γ-H2AX levels at their TSS, present lower levels of antisense transcription both by PRO-Seq and ChIP-Seq of RNA Pol II (Fig. 4G). Altogether, our data indicate that TSS genomic regions are particularly challenging to replicate in the absence of TRAIP. Moreover, although γ-H2AX signal is generally increased at the majority of TSS, it is the TSS of the first transcribed gene with high levels of antisense to sense transcription

that DNA damage most often arises at and can be detected at population level.

**Cells experience more DNA damage-generating encounters between replication and transcription upon TRAIP degradation**
As we observed that DNA damage was preferentially located at TSS following TRAIP degradation, we hypothesised that the RNA polymerase accumulated at these sites could present a barrier for the

**Fig. 3 | TRAIP is essential for S-phase progression during unperturbed cell cycle. A** TRAIP-mAID cells were arrested in G1 stage of the cell cycle and TRAIP degraded before release of cells into S-phase. Cell-cycle progression was analysed by FACS at 20 and 24 h post release. The experimental set-up (left), example of cell cycle profiles at the time of synchronisation (middle, top), examples of FACS profiles at 20 and 24 h post release (right), and quantification of cells with G2/M DNA content (mean ±SEM) over three independent experiments with two independent clones (middle, bottom). Pairwise hypothesis testing conducted using $t$ tests, with significance indicated (TRAIP-mAID C2: 20 h G1 – $p = 0.0288$, 24 h G1 – $p = 0.0171$, 20 h G2/M – $p = 0.00514$, 24 h G2/M – $p = 0.00514$; TRAIP-mAID D5: 20 h G1 – $p = 0.00825$, 24 h G1 – $p = 0.0196$, 20 h G2/M – $p = 0.0198$, 24 h G2/M – $p = 0.00517$. **B** TRAIP-mAID cells were arrested in G1 stage of the cell cycle, released into S-phase, and auxin (IAA) added in middle/late S-phase. Cell-cycle progression was analysed by FACS at 20 and 24 h post cell release into S-phase. All analyses presented as in A. **C** TRAIP-mAID cells were arrested in G2 stage of the cell cycle and TRAIP degraded before release of cells into mitosis. After release auxin (IAA) was optionally washed off 3 h after release, so that cells progressed through next S-phase with or without TRAIP. Cell-cycle progression was analysed by FACS at 24 h post release from G2 arrest. All analyses presented as in **A**. Any difference between treatment groups was determined using one-way ANOVA and post hoc testing, significance is summarised on the plot (one-way ANOVA: $p < 0.0001$; Tukey's post hoc testing, TRAIP-mAID C2: −IAA vs +IAA, $p < 0.001$; −IAA vs +IAA WASH, $p = 0.800$; +IAA vs +IAA WASH, $p < 0.001$. TRAIP-mAID D5: −IAA vs +IAA, $p = 0.00107$; −IAA vs +IAA WASH, $p = 0.898$; +IAA vs +IAA WASH, $p = 0.00554$). Source data are provided as a Source Data file.

passing replisome. To test this, we decided to inhibit the loading of the RNA Pol II at TSS using the TFIIH inhibitor triptolide[34]. Cells were arrested in G1 where TRAIP was degraded before being released into S-phase. Upon S-phase entry, cells were exposed to triptolide for 90 minutes, as it has been shown previously that short-term treatment with triptolide does not affect DNA replication progression[35] (Supplementary Fig. 10A). Strikingly, all of the DNA damage observed upon TRAIP degradation in HCT116 cells could be rescued by inhibiting RNA Pol II recruitment to the TSS by triptolide treatment (Fig. 5A). This rescue was also observed in hTERT-RPE1 cells (Fig. 5B). Importantly, we also tested whether the DNA damage specifically at TSS can be abolished by triptolide treatment. To do so, we selected four bi-directional TSS that were enriched in γ−H2AX signal upon TRAIP degradation (Fig. 4) and designed primers that could amplify them by PCR. We arrested cells in G1, released into S-phase, and treated them with triptolide as in the previous experiment. This time, however, we performed γ−H2AX ChIP followed by RT-PCR to monitor for levels of damage specifically at these genomic loci. Reassuringly, our selected TSS sites showed an increase of γ−H2AX signal after TRAIP degradation, which was rescued by triptolide treatment in three cases (Fig. 5C). In the case of the remaining TSS (Ph4B), the level of γH2AX induced after triptolide treatment alone was as high as after TRAIP degradation, so the combined treatment did not decrease the damage signal (Fig. 5C).

We then treated cells in a similar way with another transcription inhibitor DRB. DRB is a CDK9 inhibitor that leads to transcription inhibition through the accumulation of RNA Pol II at the TSS, inhibiting its progression through the gene body[36]. We could observe that DRB treatment alone created an increased level of γ-H2AX and 53BP1 foci, analogous to that of TRAIP degradation. DRB treatment did not lower the proportion of cells displaying an increased number of γ-H2AX and 53BP1 foci upon TRAIP degradation, but the effect of DRB treatment was also not additive with TRAIP degradation (Supplementary Fig. 10B). Altogether, these data suggest that the mechanism by which TRAIP is essential during S-phase is dependent on the presence of RNA Pol II specifically at the TSS.

Conflicts or collisions between the replicative helicase and transcription machinery are known to be a major source of genomic instability in otherwise unperturbed cells; through both interactions between the protein complexes themselves, or the formation of DNA:RNA hybrids (R-loops)[37]. Given the requirements of RNA Pol II bound DNA for the accumulation of DNA damage after auxin treatment, it is likely that TRAIP is required for either limiting the creation of conflicts between DNA replication and transcription or for the resolution of such encounters. We first tested therefore whether degradation of TRAIP increases levels of transcription in the cells by measuring the total level of RNA synthesis but observed no differences (Supplementary Fig. 11A). Next, we tested whether degradation of TRAIP-mAID increased the total level of RNA Pol II on chromatin, but this also was not the case (Supplementary Fig. 11B). To directly determine whether there is a higher level of replication-transcription conflicts in the absence of TRAIP, we used proximity ligation assays (PLA)

to explore any differences in the proximity of active transcription and nascent DNA. Cells were arrested in G1 where TRAIP was degraded before being released into S-phase. Approximately 12.5 hours following release when cells were in early S-phase (known to have high levels of both transcription and replication and when our γ−H2AX hotspots are replicated), the thymidine analogue EdU was added for 20 minutes. Cells were then harvested, and PLA assays were used to detect interactions between active transcription (phosphorylated RNA Pol II) and nascent replicated DNA (Fig. 5D). Intriguingly, we observed an increase in the amount of active RNA Pol II present in proximity to nascent DNA following TRAIP degradation, suggesting that TRAIP is indeed important for resolving replication-transcription encounters on chromatin and maintaining fork progression. To confirm this result, we also used the PLA assay to determine the proximity of RNA Pol II to components of the replisome: PCNA, and AND-1, with similar results: an absence of TRAIP during S-phase led to an increase in incidences of RNA Pol II and replisome proximity (Fig. 5E).

## TRAIP depletion does not lead to DNA damage during S-phase in absence of transcription

To further explore the importance of TRAIP for the regulation of replication-transcription collisions we next turned to the *X. laevis* egg extract model system, which can support robust DNA replication activity in the absence of any transcription. During early embryogenesis in *Xenopus* embryos 12 cleavage cell cycles are achieved without transcription and only restricted protein translation. Most required factors for DNA replication and cell division are accumulated in the egg. The significant level of gene transcription is induced in the embryo only at the stage of midblastula transition. As *Xenopus* egg extract is derived from *Xenopus* eggs and the replicated DNA substrate is demembranated *Xenopus* sperm, the replication observed resembles embryonic replication during early cleavage divisions in the absence of transcription.

We raised antibodies against *X.l.*TRAIP (Supplementary Fig. 12A) and using this antibody we immunodepleted TRAIP from *Xenopus* egg extract to less than 10% of its original quantity (Fig. 6A). We could observe that although a lack of TRAIP in the extract did not affect its ability to synthesise DNA during S-phase, as previously reported[23], it did inhibit mitotic unloading of post-termination replisomes, which is a known function of TRAIP in this system (Supplementary Fig. 12B). This verified that our immunodepleted extract was indeed devoid of TRAIP's activity. Given our findings in the mammalian system, we next examined whether we could observe any signs of creation of DNA damage or checkpoint activation during S-phase without TRAIP in *Xenopus* egg extract system lacking transcriptional activity. We could detect no increase of γ-H2AX signal on chromatin upon TRAIP depletion, despite the DNA damage signalling cascade being functional in TRAIP-depleted extract, as treatment with restriction enzyme (EcoRI), which induces double-strand breaks, could activate a robust γ−H2AX signal on chromatin in both IgG- and TRAIP-depleted extracts (Fig. 6B). Similarly, we could observe no induction of phosphorylated Chk1 in

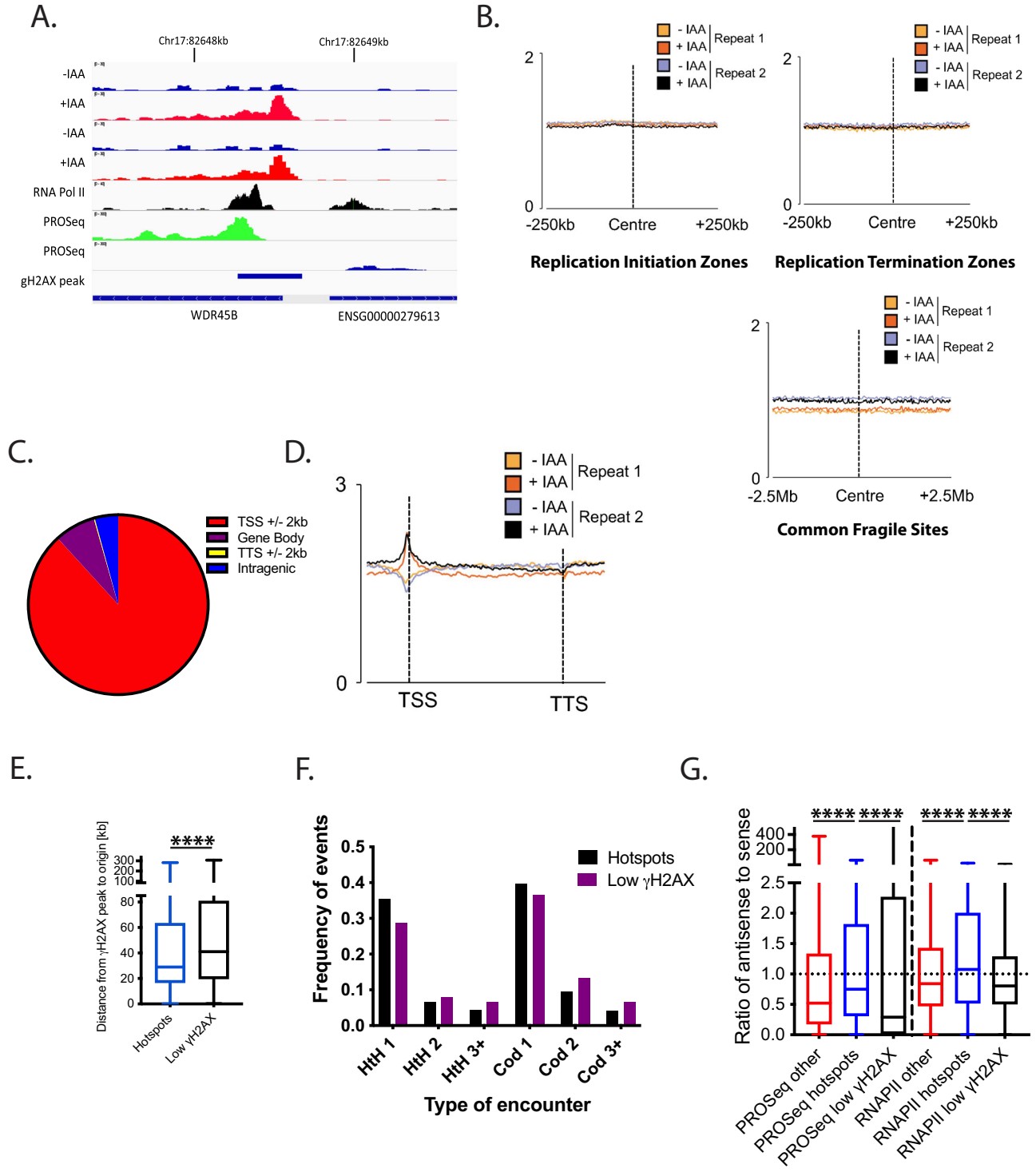

the nucleoplasm in the absence of TRAIP, while such a signal was readily induced by treatment of replicating extract with inhibitor of replicative polymerases (aphidicolin) and inhibited by ATM/ATR inhibitor (caffeine) (Fig. 6C). This further supports our hypothesis that TRAIP works in S-phase to protect genome stability by resolving conflicts between replication and transcription machineries; in the absence of transcription, such DNA damage does not occur.

**TRAIP ubiquitylation substrates are likely unloaded by UBXD7-p97 complex**
We next wanted to determine the mechanism by which TRAIP acts to resolve replication-transcription encounters. Previous research has

indicated that the formation of R-loops is often the underlying cause of replication-transcription conflicts[38] and R-loops accumulating at the TSS can stimulate TSS-associated transcription[39]. We tested therefore whether we can observe the accumulation of R-loops in response to TRAIP-mAID degradation, as detected by S9.6 antibody, but did not see their accumulation (Supplementary Fig. 11C). Moreover, the DNA damage induced by TRAIP-mAID degradation could not be rescued by overexpression of RNAseH1, which can degrade RNA:DNA hybrids (Supplementary Fig. 11D). Altogether, the DNA damage created in absence of TRAIP due to replication-transcription collisions is likely not driven through resolution of R-loops on chromatin.

**Fig. 4 | Absence of TRAIP during S-phase leads to generation of DNA damage at transcription start sites (TSS). A** An example of TRAIP degradation induced γH2AX peak over two independent repeats of γH2AX ChIP-seq experiments. Transcription direction and position of accumulation of RNA Pol II[34] and of PRO-Seq[33] around the promoter are also presented. **B** γH2AX ChIP signal from two independent experiments was compared with positions of replication initiation and termination zones as mapped by[30], or common fragile sites as mapped by[32]. Each graph is centred around the centre of initiation/termination/fragile site. **C** Graphical representation of the position of the called conserved γH2AX peaks (damage hotspots), with 88.24% of γH2AX peaks in proximity of TSS. **D** Average binned metagene profile for γH2AX levels normalised to H2AX at transcribed genes in HCT116 cells, from TSS to transcription termination sites for the two repeat experiments. **E** Distance between the γH2AX peaks at TSS at hotspots and the TSS of genes with the lowest fold change in γH2AX levels and origins of replication identified in Koyanagi et al., 2021. The positions of TSS with called common γH2AX

peaks are significantly closer to the origin of replication than the distance between TSS and origin of replication in genes that do not show increased γH2AX signal upon TRAIP degradation. Box whiskers plots with line at the median; Mann–Whitney $t$ test; ****$p$ value < 0.0001. **F** The proportion of 1st, 2nd, and further on (3+) transcribed genes encountered by replication forks emerging from the closest replication origin, presented for presented for "hotspots" = genes with reproducible γH2AX peak in two experimental repeats; and "Low γH2AX" = genes with no increase in γH2AX levels. Chi-square $p$ value < 0.05. **G** Quantification of the ratio between antisense- and sense transcription at TSS ± 1 kb of hotspots, all other transcribed genes, and transcribed genes with low levels of γH2AX increase following TRAIP depletion. PRO-Seq strand-specific signal[33] and RNA Pol II chromatin immuno-precipitation sequencing[34] was quantified −1000 bp to TSS for antisense transcription, and TSS to 1000 bp for the sense transcription. Box whiskers plots with line at the median; Mann–Whitney $t$ test; ****$p$ value < 0.0001. Source data are provided as a Source Data file.

TRAIP was shown previously to be able to ubiquitylate for removal a DPC barrier in front of the replisome[18], but also to ubiquitylate replisomes that converge at DNA inter-strand crosslink (ICL) leading to their removal from chromatin[13]. Moreover, RNA Pol II is known to be ubiquitylated by a number of ubiquitin ligases in order to remove it from chromatin (reviewed in ref. 40). It is possible therefore that TRAIP can ubiquitylate either RNA Pol II machinery, which is presenting to replication fork as a barrier, or the replisome itself. To investigate potential mechanism of TRAIP action we first explored any interactions between TRAIP and RNA Pol II in the cells, to see if TRAIP is present at the right place to fulfil this job. In order to do so, we took advantage of the Clover tag in TRAIP-mAC and looked for a proximity signal arising from TRAIP-mAC and RNA Pol II in the cells, using GFP antibody to detect TRAIP-mAC using the PLA technique. We could indeed detect such a proximity that was lost upon auxin addition and TRAIP-mAC degradation (Fig. 7A). As TRAIP was also reported to be enriched at replisomes in human cells[7,14,15], it has, therefore, the opportunity to act on either RNA Pol II or the replisome to resolve the conflict between these machineries and avoid DNA damage.

Both, ubiquitylated RNA Pol II[41] and the replisome[13] are removed from chromatin with assistance of p97 segregase. We aimed therefore to determine whether the TRAIP-driven ubiquitylation at transcription-replication collisions would be followed by p97 extraction of ubiquitylated complexes. However, we and others have shown that inhibition of p97 with small molecule inhibitors leads to the generation of far more DNA damage and inhibition of S-phase progression than in the case of TRAIP degradation[42–44]. We decided therefore to look for a cofactor of p97 that may direct p97 to TRAIP-ubiquitylated substrates at replication-transcription encounters. We started with the two most likely candidates: SPRTN, which assists p97 with removal from chromatin of DPCs ubiquitylated by TRAIP[18,45] and UBXD7, which was shown to facilitate replisome onloading during replication termination[46,47] and is also a human homologue of the *S.cerevisiae* Ubx5 cofactor facilitating not only unloading of RNA Pol II from chromatin in budding yeast, but also assisting Wss1 protease and Cdc48 (*S.cerevisiae* p97) at DPCs[41,48]. Interestingly, siRNA down-regulation of SPRTN did not lead to generation of DNA damage (γ−H2AX and 53BP1 foci) in otherwise unperturbed S-phase. Conversely, siUBXD7 did lead to an accumulation of DNA damage repair foci in S-phase to a similar level as after TRAIP-mAID degradation, and combining siUBXD7 and TRAIP degradation did not further increase the level of detected DNA damage response (γ−H2AX and 53BP1 foci), suggesting that these two factors are epistatic and act in the same pathway (Fig. 7D).

## Discussion
Our work leads us to propose a model of TRAIP activity during S-phase. TSS with bi-directional transcription present a uniquely difficult problem for replication, as replication forks are likely to approach

transcription machinery in a head-to-head orientation on both sites of such TSS. These sites are more likely to result in replication-transcription collisions, which can be resolved by TRAIP ubiquitin ligase activity. TRAIP can ubiquitylate either RNA Pol II or the replisome in such situations, leading to unloading of the problematic machinery by the p97-UBXD7 complex and resolution of the conflicts without generation of DNA damage (Fig. 8).

Over the years, much evidence has accumulated indicating that TRAIP ubiquitin ligase plays wide-ranging roles in the maintenance of genomic integrity. TRAIP-depleted cells were shown previously to display a diverse spectrum of phenotypes: gross chromosomal re-arrangements, problems with DNA damage response activation, micronuclei, and accumulation in G2 stage of the cell cycle[6–8]. The functions of TRAIP were studied upon specific stimuli e.g., induction of double-strand breaks, induction of inter-strand crosslinks, stimulation of MiDAS, or inhibition of replicative polymerases etc.[7,13,19,22]. However, the primary consequence of TRAIP depletion, the essential function of TRAIP for cell viability in unperturbed conditions, was not known.

Here, we have taken advantage of the degron system to rapidly degrade TRAIP within cells upon auxin addition, to dissect the importance of various functions of TRAIP in different stages of the cell cycle. TRAIP-deficient human cells have been shown to proliferate slowly even in the absence of exogenous DNA damage and the reduction of cellular proliferation during development is a likely cause of microcephalic dwarfism observed in patients with TRAIP mutations[6,49]. Here, for the first time we have determined the mechanism of inhibition of cell proliferation in TRAIP-depleted cells. Firstly, we have shown that upon degradation of TRAIP, cells show signs of spontaneous DNA damage, leading to ATR checkpoint activation, G2 cell cycle arrest, and withdrawal from the cell cycle into senescence. Senescence is classically induced by telomere erosion in cells as the organism ages, due to activation of DDR. However, other stresses that engage DDR, such as exposure to oxidants, γ-irradiation, UVB light, or DNA-damaging chemotherapies can all induce senescence (reviewed in ref. 50). Many of these DDR stimuli can also induce apoptosis. It has been suggested that the level of stress can influence the choice between senescence and apoptosis, with senescence being induced by less severe damage[50,51]. We can observe a low level of DNA damage foci induced by degradation of TRAIP, which is likely the reason for this cell fate choice. Senescence was also classically mechanistically defined as an irreversible cell cycle arrest in G1 phase (G0). However, many senescent cells in our body have 4 N DNA content[52] and several studies showed that p21 also mediates permanent DNA damage-induced cell cycle arrest in G2, by inhibiting mitotic CDK complexes and pRb phosphorylation (reviewed in[28]). Interestingly, this choice of a cease of proliferation by senescence may have additional consequences for TRAIP mutation-driven disease onset. Senescent cells are not eliminated but stably viable and can influence neighbouring cells through secreted soluble factors (senescence-

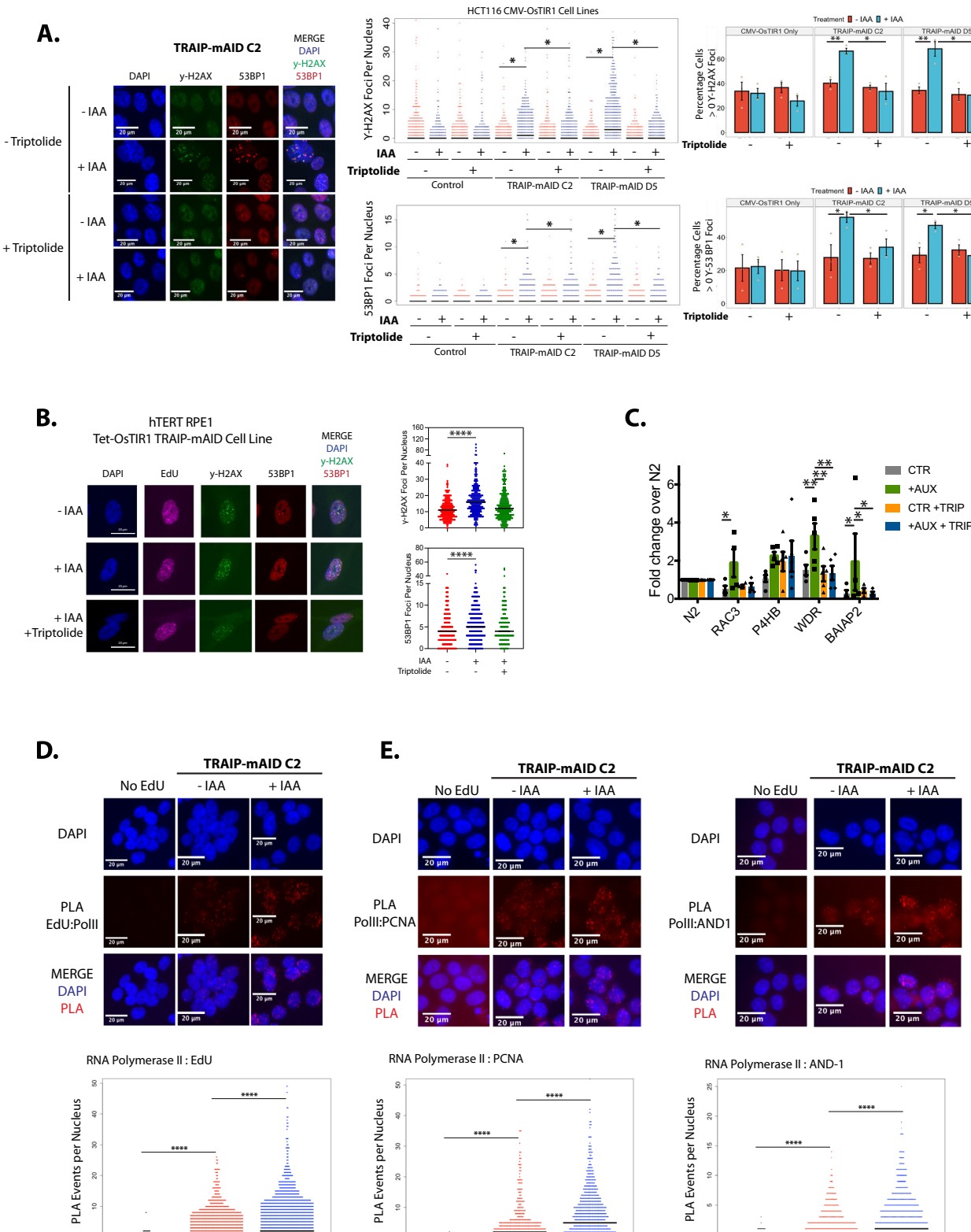

associated secretory phenotype, SASP). SASP attracts inflammatory cells to eliminate senescent cells but has been also associated with tissue and organ deterioration[50].

TRAIP has been shown over the years to have many functions in mitosis. TRAIP can unload rogue replisomes during mitosis[20–22], is important for execution of MiDAS[21–23,53], and regulates the spindle assembly checkpoint[10,25]. However, the insults to genome stability

generated by cells progressing through mitosis without TRAIP are not responsible for the perturbed cell proliferation and G2 arrest. Instead, we conclude that the functions of TRAIP during S-phase are of utmost importance for cell viability. This is not to say that TRAIP does not have roles in mitosis. Indeed, we can observe delayed progression through mitosis when TRAIP was degraded specifically before onset of mitosis. It is also likely that a continuous lack of TRAIP from S-phase and

**Fig. 5 | DNA damage induced in S-phase by TRAIP degradation depends on presence of transcription machinery. A** HCT116 TRAIP-mAID cells were arrested in G1 stage of the cell cycle and TRAIP degraded before release of cells into S-phase. Upon S-phase entry cells were exposed to triptolide for 90 min. γH2AX and 53BP1 foci induced were visualised by immunofluorescence and quantified. Example pictures (left) and quantification of three independent experiments is presented as quantification of number of γH2AX and 53BP1 foci per nucleus (middle) or percentage of positive cells (>0 foci) in population (right) (mean ±SEM)). Differences between conditions was identified using *t* tests (yH2AX: TRAIP-mAID C2 - Triptolide: *p* = 0.00622; TRAIP-mAID D5 - Triptolide: *p* = 0.00798; TRAIP-mAID C2 + IAA ±Triptolide: *p* = 0.0297; TRAIP-mAID D5 + IAA ±Triptolide: *p* = 0.0318). 53BP1: TRAIP-mAID C2 - Triptolide: *p* = 0.0424; TRAIP-mAID D5 - Triptolide: *p* = 0.0474; TRAIP-mAID C2 + IAA ±Triptolide: *p* = 0.0286; TRAIP-mAID D5 + IAA ±Triptolide: *p* = 0.03709). **B** TRAIP was degraded in hTERT-RPE1 TRAIP-mAID cells for 24 h. In the last 90 min, cells were optionally exposed to triptolide and in the last 20 min, cells pulsed with EdU. γ–H2AX, 53BP1 foci, and EdU incorporation were visualised by immunofluorescence and quantified. Example pictures (left) and quantification of 2–3 independent experiments is presented as number of γ–H2AX and 53BP1 foci

per EdU-positive nucleus. Mann–Whitney *t* test; ****p* value < 0.0001. **C** HCT116 TRAIP-mAID cells were treated as in **A** but instead of DNA damage foci analysis, γH2AX ChIP was conducted followed by RT-PCR to detect indicated TSS. N2 is a amplifying a fragment from gene desert on chromosome 13. Mean of *n* ≥ 3 with SEM. T.test per gene per condition: PH4B ctr vs IAA *p* = 0.034, WDR ctr vs IAA *p* = 0.018, WDR IAA vs IAA+Trip *p* = 0.016. **D** TRAIP-mAID cells were arrested in G1 stage of the cell cycle and TRAIP degraded before release of cells into S-phase. In early S-phase EdU was added for 20 min and proximity between incorporated EdU and RNA Pol II visualised by PLA. Example pictures (top) and quantification all PLA events per nucleus are presented (bottom). Mann–Whitney *t* test; ****p* value < 0.0001 (*n* = 3). **E** PLA between PCNA and RNA Pol II (left) and AND-1 and RNA Pol II (right) after optional treatment with auxin. Experiments were carried out analogously to that detailed above; cells were arrested in G1 where TRAIP was degraded through IAA treatment, before being released into S-phase, and samples taken ~12.5 hours post release. Example pictures (top) and quantification of all PLA events per nucleus are presented (bottom). Mann–Whitney *t* test; ****p* value < 0.0001 (*n* = 3). Source data are provided as a Source Data file.

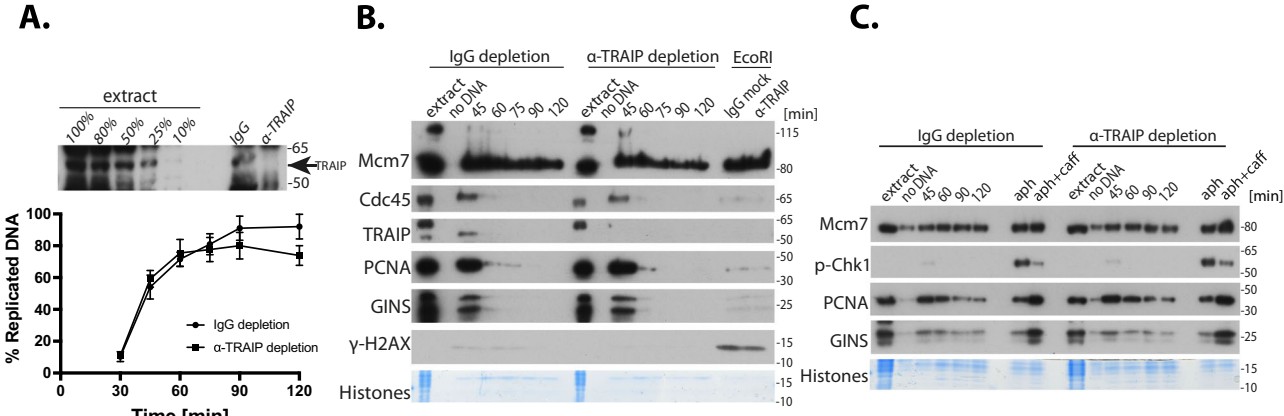

**Fig. 6 | Depletion of TRAIP does not induce S-phase checkpoint activity in** *Xenopus laevis* **egg extract. A** TRAIP was immunodepleted from *Xenopus* egg extract to less than 10% as judged by immunoblotting of depleted extract (top). The ability of TRAIP-depleted extract to synthesise DNA was established by measuring α³²P-dATP incorporation into newly synthesised DNA and compared with replication of non-specific IgG-depleted extract. **B** DNA replication reaction was established in non-specific IgG- and TRAIP-depleted egg extracts and chromatin isolated at indicated timepoints during replication reaction. Chromatin samples were analysed by western blotting with indicated antibodies. A "no DNA" sample shows

chromatin specificity of the signal. Histones stained with Coomassie serve as a loading control. Samples of both extracts treated with restriction enzyme EcoRI to induce double-strand breaks (isolated at 60 min of reaction) serve as positive control for γH2AX signal. **C** DNA replication was established as in **B** but instead of chromatin whole nuclei containing nucleoplasm were isolated to measure level of active phosphorylated Chk1. Samples of both extracts treated with the DNA Polymerase inhibitor aphidicolin serve as positive controls for Chk1 activation, while samples treated with aphidicolin and caffeine indicate ATM/ATR dependence of these signals. Source data are provided as a Source Data file.

through mitosis, exacerbates the problems experienced by cells depleted of TRAIP in S-phase. Some cells undergoing DNA damage-induced G2 senescence have been shown with time to be able to escape the G2/M checkpoint, progress through mitosis, and arrest in the subsequent G1 phase. This progression can also happen without chromosome segregation (mitotic bypass) leading to an accumulation of tetraploid G1 cells[28]. In either case, the replicative problems generated during S-phase due to the absence of TRAIP, could present a substrate for TRAIP activity during mitosis and mitotic functions of TRAIP could reduce the overall genomic instability generated by TRAIP depletion. Unfortunately, we are unable to test this possibility due to the time required to resynthesise TRAIP, which is incompatible with S-phase without TRAIP and following mitosis with TRAIP.

TRAIP has been shown previously to be important for the cellular response to DNA replication stress upon different types of insults[6–8]. It has also been suggested to be a master regulator of ICL repair during DNA replication, as its activity promotes two alternative pathways of ICL repair in *X. laevis* egg extract[13]. However, patients with TRAIP mutations display dwarfism rather than the classical Fanconi anaemia clinical outcome, which is characteristic for mutations within ICL

repair factors[6,54]. Instead, our results indicate that without exogenous sources of replication stress, the major endogenous source of replication fork impediments that require TRAIP activity is in encountering transcription machinery. In the absence of TRAIP during S-phase in human cells, we observe increased levels of persisting DNA replication-transcription encounters. We can also detect enrichment of the DNA damage response at TSS sites, where replication forks likely collide with RNA Pol II. Importantly, this damage can be completely rescued by the temporary removal of RNA Pol II from chromatin. Moreover, in the *Xenopus* egg extract model system where efficient DNA replication can be established without the presence of concurrent transcription, TRAIP is not essential for the completion of S-phase, nor can we detect any signs of DNA damage or checkpoint activation.

In human cells, coexistence of DNA transcription and DNA replication is a well-established potential source of endogenous replication stress. It is often exacerbated by oncogenic deregulations stimulating transcription, whilst simultaneously promoting premature entry into S-phase, leading to a higher probability of interference between the two processes[35,55]. The role of TRAIP in the resolution of such conflicts is consistent with the observation that TRAIP-deficient cells show fork

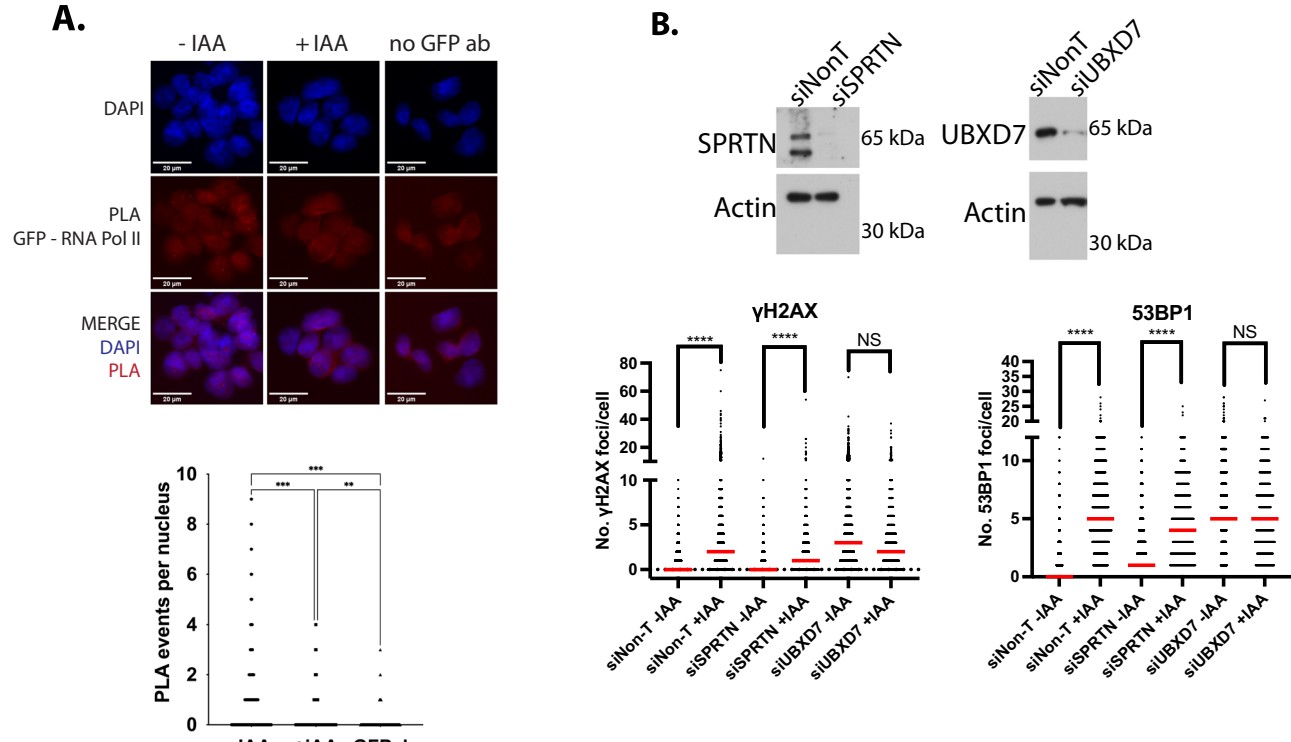

**Fig. 7 | TRAIP facilitates resolution of replication transcription encounters.**
**A** PLA assay between TRAIP-mAC (GFP antibodies) and RNA Pol II. Example pictures (top) and quantification of all PLA events per nucleus are presented (bottom).
**B** SPRTN or UBXD7 were downregulated with siRNA in HCT116 TRAIP-mAID cells and the protein level verified by western blot (left). TRAIP-mAID was degraded for 24 h and γH2AX or 53BP1 foci were visualised and quantified in cells in S-phase (right) (*n* = 3). Mann–Whitney *t* test; ****p value < 0.0001. Source data are provided as a Source Data file.

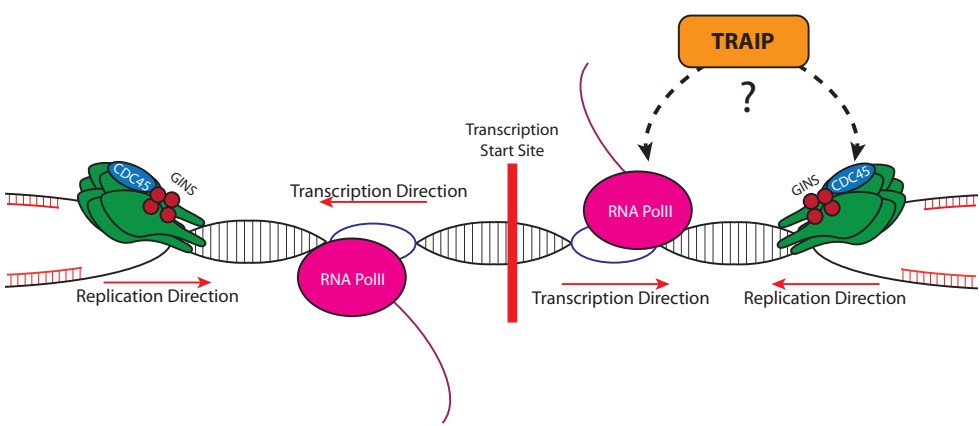

**Fig. 8 | Model of TRAIP activity during S-phase.** During unperturbed DNA replication the bi-directional transcription start sites (TSS) represent a difficult to replicate region, as replication forks approaching such TSS from either direction, encounter transcription machinery in a head-to-head orientation. Such collisions can lead to genomic instability if not resolved. TRAIP facilitates resolution of these replication-transcription encounters through its ubiquitin ligase activity directed either towards the replisome or the transcription machinery.

asymmetry in the DNA fibre assays, suggesting site-specific stalling of one of the forks emanating from the same origin of replication[7].

What is the mechanism by which TRAIP may promote resolution of replication-transcription collisions? Our analysis of the hotspots of DNA damage created in the absence of TRAIP suggests that the problems most often arise when the newly established replication fork encounters the first gene promoter where RNA Pol II has accumulated. Moreover, the hotspot sites represent particularly crowded sites, with high levels of transcription moving in both directions from the promoter. Despite the orientation of annotated gene transcription being equally often in head-to-head or codirectional with the direction of progression of the replication fork, the equal levels of sense to antisense transcription at hotspot TSS suggest that the fork will ultimately encounter the transcription machinery in a head-to-head orientation. These bi-directional transcription units may therefore represent a particularly difficult impediment for replication fork progression. Indeed, sites with high levels of antisense transcription have already been identified as hotspots for transcription-replication interactions and for G2/M DNA synthesis[38,56]. Previous research has also identified that TSS-associated transcription is stimulated by the presence of R-loops accumulating at the TSS[39] and the propensity to form R-loops is a feature of transcription-replication interaction hotspots[38].

However, when we assessed R-loop levels using the R-loop specific antibody S9.6 by immunofluorescence, we did not identify an increase in cells depleted of TRAIP, and overexpression of RNAseH1 did not rescue the DNA damage generated in S-phase without TRAIP. These results suggest that TRAIP depletion is not leading to a global increase in R-loop levels in cells, and their resolution is not important for TRAIP-driven mechanisms.

The transcription machinery encountered by the replication fork very much resembles a DNA-protein barrier and TRAIP has been shown previously to facilitate replisome bypass of DPC (protein covalently crosslinked to DNA by aldehydes or chemotherapeutics)[18]. In this situation, TRAIP stimulates the bypass of the DPC by the replication machinery and also ubiquitylates DPC to stimulate its degradation by SPRTN and the proteasome[18,57]. It is therefore likely that TRAIP could act similarly when encountering tightly associated RNA Pol II. It is well-established that removal of RNA Pol II from chromatin in response to DNA damage is driven by ubiquitylation of RNA Pol II and subsequent unfolding and unloading by p97 segregase[40]. We have shown here that TRAIP and RNA Pol II can be found in close proximity in S-phase nuclei. However, more research is needed to establish RNA Pol II as TRAIP substrate. Alternatively, the replisomes themselves could be the substrate of TRAIP ubiquitylation activity. Whatever the substrate, our data suggest that p97 together with UBXD7 is likely to unload ubiquitylated complexes from chromatin following TRAIP ubiquitylation.

Finally, downregulation or immunodepletion of TRAIP in human cells and *Xenopus* egg extract system was reported previously to lead to chromosomal instability and re-arrangements, which are a hallmark of cancer development[7,8,23]. Notably, however, TRAIP-deficient patients were not reported to characterise with cancer predisposition, similarly to other Seckel syndrome patients[6,49]. It is possible therefore that the G2 arrest and senescence we observe upon degradation of TRAIP is not compatible with increased proliferation and tumour development. Inhibition of TRAIP could therefore be significantly detrimental to cancerous cells.

## Methods

### Plasmid construction

DONOR and CRISPR plasmids were constructed for the generation of conditional Auxin-Inducible Degron Cell lines as detailed in Natsume et al.[26]. CRISPR-Cas9 was expressed using the pX330-U6-Chimeric_BB-CBh-hSpCas9 plasmid[58] (Addgene #42230) and targeted the C-terminus coding region containing the stop codon (5′-CTCACTGTTCTCACGACCAC-3′). Donor plasmids were generated following a published protocol[27]. Briefly, we cloned homology arms upstream and downstream of the CRISPR target region to pBluescript[26,27] (~500 bp each). After inverse PCR, a cassette containing mAID or mAID-Clover (mAC) with a selection marker was cloned to make a donor plasmid. Two donor plasmids were created: *TRAIP*-mAID and *TRAIP*-mAC, containing either Hygromycin (HCT116) or Neomycin (hTERT-RPE1) resistance for selection.

### Generation of TRAIP conditional degron cell lines

HCT116 cell lines constitutively expressing the *Oryza sativa* auxin-sensitive F-box protein (OsTIR1)[26] were transfected with the previously described DONOR and CRISPR plasmids. Parental HCT116 cell lines were seeded to individual wells of a six-well plate at a final working concentration of $1 \times 10^5$ cells/ml. Transfection was conducted 2 days following plating using Fugene HD, OPTI-MEM, 200 ng/µl DONOR plasmid, and 200 ng/µl CRISPR plasmid. Parental hTERT-RPE1 cell lines were diluted to a working concentration of $1 \times 10^6$ cells/ml, washed with PBS, and re-suspended in Resuspension Buffer R (Neon Transfection System, Thermofisher Scientific). The re-suspended cell reaction was supplemented with both DONOR and CRISPR plasmids (1500 ng total) and electroporated 3× at 1350 V for 20 ms, 2 pulses per reaction. The resulting electroporated cell suspension was transferred

to individual wells of a six-well plate and incubated for 2–3 days before being subject to the selection procedure below. Control reactions were transfected with DONOR plasmids only. Approximately 24 hours later for HCT116 cells or 72 hours later for hTERT-RPE1 cells, the transfected cultures were collected and diluted at different concentrations in 10 cm dishes. The selection was carried out 24 hours after dilution (100 µg/ml Hygromycin or 1 mg/ml G418) and maintained continuously for 11-13 days. Once sufficient colonies were observed, 48 from each condition were isolated, grown in individual wells of a 96-well plate until confluent, and screened for bi-allelic gene insertion using genomic PCR. Adherent cells were washed twice with Dulbecco's PBS (−) (D-PBS (−)) and treated with DirectPCR Lysis Reagent (Viagen: 0.5× DirectPCR Lysis Reagent, 20 mg/ml proteinase K) overnight at 55 °C. Proteinase K activity was stopped through incubation of the 96-well plate at 85 °C for 90 minutes in humid conditions. Genomic PCR was carried out using GoTaq HotStart Green Master Mix (Promega), as per manufacturer's instruction (*TRAIP* F: AGATGGGTGAGTGTGGCTTC; *TRAIP* R: GCTGCAGTGATCT-CATTCTTTCT; mAID: ATCTTTAGGACAAGCACTCTTCTCC). Bi-allelic gene insertion was first screened using the Microchip Electrophoresis System (Multi-NA; Shimadzu), and any subsequent bi-allelic tag status confirmed using DNA gel electrophoresis with 2% agarose gels. HCT116 cells were maintained in McCoys's 5 A medium supplemented with 10% Fetal Bovine Serum, 2 mM L-glutamine, 100 U/ml Penicillin, and 100 µg/ml Streptomycin; hTERT-RPE1 cells were maintained in DMEM-F12 medium supplemented analogously to McCoy's 5 A medium using Tetracycline-free Fetal Bovine Serum. All cultures were grown at 37 °C with 5% $CO_2$.

### Generation of RNAse HI-GFP stable cell lines

HCT116 CMV-OsTIR1 TRAIP-mAID cell lines were further modified through viral transduction to introduce the Dox-inducible over-expression of GFP-tagged RNAse HI. RNAse HI-GFP plasmids were a kind gift from Dr Kienan Savage[59]. Virus production was carried out using HEK293T cells, cultured in DMEM media supplemented as detailed for DMEM-F12. To produce virus, cells were harvested, re-suspended in IMDM media supplemented as detailed, and seeded at a working concentration of $3 \times 10^5$ cells/ml on $2 \times 15$ cm tissue culture dishes. Approximately 24 hours later, cells were transfected as detailed. A total of 25 µg of RNAse HI-GFP plasmid was combined with 16.25 µg of packaging plasmid and 9 µg of envelope plasmid. Plasmids were diluted appropriately in 0.1× TE buffer and distilled $H_2O$ before being supplemented with 2.5 M $CaCl_2$. Finally, 2× HBS was added to a final concentration of 1× and the transfection mixture was added to the cells. Precisely 16 hours following transfection the media was changed to fresh supplemented IMDM. Approximately 24 hours later, the cell media was collected, filtered using a 0.22 µM syringe filter, and centrifuged to remove cell debris (400 × *g*, 5 minutes). The resulting virus suspension was aliquoted and stored at −80°C until use. To transduce cells, HCT116 CMV-OsTIR1 TRAIP-mAID clones were seeded onto six-well plates at a working concentration of $2 \times 10^5$ cells/ml. The virus was added dropwise to the cells and incubated for 24 hours; when the media was replaced with pre-warmed supplemented McCoy's 5 A. Approximately 24 hours later, cells were harvested and plated onto 10 cm dishes at a 1–200 dilution and treated with 10 µg/ml Blasticidin. Cells were incubated with the appropriate antibiotic for 7–10 days, changing the media every 3 days. RNAse HI expression was confirmed through GFP expression under a fluorescent microscope.

### Generation of TRAIP rescue cell lines

HCT116 CMV-OsTIR1 TRAIP-mAID cell lines were modified to introduce constitutively expressed wild-type, C7A/C10A, or W37A mutant TRAIP through retrovirus transduction. Retrovirus plasmid constructs containing the *TRAIP* sequences were a kind gift from Prof. Andrew Jackson[6]. Virus production was carried out using HEK293T cells, grown

as detailed previously. Briefly, HEK293T cells were seeded onto 10 cm tissue culture plates and grown until approximately 40-50% confluent. The transfection plasmids were prepared as follows: 300 µl OptiMEM reduced serum media, 6 µg *TRAIP* construct, 3.8 µg Clonetech GAG/POL plasmid, 2.2 µg Clonetech VSVG plasmid. This mix was combined with 24 µl PEI pre-prepared in 300 µl OptiMEM and incubated at room temperature for 20 minutes. The transfection mix was added dropwise to the HEK293T cells and cells incubated overnight. Approximately 16 hours later the media was replaced with fresh, pre-warmed media and incubated for a further 24 hours. Following this, medium was collected and passed through a 0.22 µM filter to remove cell debris. Virus was stored at −80°C until use. Parental cell lines were transduced as detailed previously, analogously to lentivirus.

### Detecting TRAIP protein levels

Immunoblotting was used to detect both TRAIP band shifts corresponding to the inserted tag size, and protein degradation. Whole-cell extracts were generated by re-suspending harvested cell pellets in UTB extraction buffer (8 M Urea, 50 mM TRIS-HCl, 150 mM B-Mercaptoethanol) for 10 minutes on ice. The cell lysate was then sonicated (BioRuptor: 30 sec on, 30 sec off, 5 min cycles, medium power) and centrifuged at $14,000 \times g$ for 20 minutes to separate soluble and insoluble fractions. The soluble fraction was collected and mixed with 4× SDS-PAGE Loading Buffer (NuPAGE) to a final concentration of 1×. Approximately 50 µg of total protein content was run-on 4–12% gradient SDS-PAGE gels (Invitrogen) and transferred to Nitrocellulose membranes for 90 minutes at 80 V. Transferred membranes were blocked (5% Milk in TBST) for 1 hour before being incubated in primary antibody at 4 °C overnight. Membranes were washed 3 × 10 minutes in TBST and incubated in secondary antibody for 2 hours at room temperature. The membranes were then washed 3 × 10 minutes in TBST before being developed using ECL detection spray (Advansta WesternBright). Antibodies used for immunoblotting were as follows: Anti-TRAIP antibodies were kindly provided by professor N. Mailand[7] and used 1:300 in 5% Milk in TBST. Anti-β Actin loading controls used 1:5000 in 5% BSA in TBST (C4 anti-Actin HRP Santa Cruz).

### Drug treatments

TRAIP degradation in HCT116 cells was induced through the addition of 500 µM Indole-3 acetic acid (IAA) to cell media, subsequently diluted 1:5 in the existing growth media to provide a final working concentration of 100 µM. Protein degradation in hTERT-RPE1 cell lines was carried out using a final working concentration of 25 µM IAA, prepared analogously to that described for HCT116 cells. The respective working concentrations of other inhibitor treatments used are described: 10 µM ATM inhibitor (Stratech KU-55933), 4 µM ATR inhibitor (Stratech AZD6738), 5 µM MK-1775 (Stratech), 100 nM Triptolide (Sigma-Aldrich T3652), 100 µM DRB (Sigma-Aldrich D1916), 0.4 µM Aphidicolin (A0781, Sigma).

### Cell viability assays

For colony-forming assays, cells were diluted to a working concentration of 1000 cells/ml and plated to individual wells of a six-well plate at different seeding concentrations (100, 250, 500, 750, 1000 cells/well). For assays using hTERT-RPE1 cell lines, 500 cells were seeded onto 10 cm tissue culture dishes. HCT116 cell lines were treated with Auxin (IAA) 24 hours after plating whilst hTERT-RPE1 cells were first treated with 100 ng/ml Doxycycline 24 hours after plating, and IAA added the following day (48 hours after plating). Cell media containing Doxycycline was refreshed every 3–4 days. All cells were incubated until sufficient colony formation was observed (~10–14 days). At this point, the cell media was removed, and colonies stained using Methylene Blue staining (2% methylene blue in 50% ethanol) buffer for 5 minutes at room temperature. Colonies were rinsed with $H_2O$ and dried overnight. For quantification, the percentage colony forming efficiency was calculated by dividing the number of counted colonies by the original number of cells plated, allowing comparison between different seeding concentrations. To then allow comparison between different cell lines, the calculated percentage colony forming efficiency was normalised to a control cell line (Untreated HCT116 CMV-OsTIR1 only or hTERT-RPE1 Tet-OsTIR1 only; Efficiency in test clone/Efficiency in Control). For cell proliferation assays, cells were diluted to $1 \times 10^4$ cells/ml and plated into either 60-mm dishes for HCT116 cell lines or 10 cm dishes for hTERT-RPE1 degron cell lines. Cells were treated with IAA and Dox as appropriate as detailed above, and cells grown for a further 72 hours. At each timepoint (24, 48, 72 hrs post auxin treatment) cells were harvested and the total cell numbers counted using a COUNTESS cell counter. Cell counts were normalised to the seeding concentration. In addition, following cell counting, the remainder of the samples were prepared for Flow Cytometry as detailed.

### Cell death assays

Cell senescence was detected using the Cell Meter Senescence Activity Assay (AAT Bioquest). Cells were plated for proliferation curves as described. At each timepoint, cells were harvested and washed with PBS. Cell pellets were re-suspended in Xite Green β-D-galactopyranoside solution for 45 minutes at 37 °C. Stained cells were washed again in PBS, re-suspended in 500 µl Assay Buffer, and analysed using a Flow Cytometer. Senescent cells were detected using the FITC 488 channel, with unstained controls utilised to distinguish between positive and negative cell populations. Apoptotic cells were detected using the Annexin V Apoptosis Detection Kit (Invitrogen). Cells were grown and harvested as detailed. The resulting pellet was re-suspended in 1× binding buffer supplemented with AF488 fluorochrome coated Annexin V. Samples were incubated for 15 minutes at room temperature and washed in Binding Buffer. Stained cells were then re-suspended in 200 µl Binding buffer supplemented with 50 µg/ml Propidium Iodide and 50 µg/ml RNase A. Cells analysed using a flow cytometer, with apoptotic cells detected using the FITC 488 channel.

### Cell synchronisation

For G1 cell synchronisation, asynchronous cells were treated with 20 µM Lovastatin (Acros Organics) for 24 hours. To then release cells from the G1 arrest, cells were washed 3× in pre-warmed growth media before fresh media supplemented with 2 mM Mevalonic Acid (Sigma-Aldrich) was added. Cells entered S-phase ~12 hours post release. To arrest cells in G2 stage of the cell cycle, asynchronous cells were treated with 9 µM RO-3306 (Merck Life Sciences) for 16 hours. G2 release was achieved through washing the cells 3× in pre-warmed media before fresh growth media was added. Released cells enter mitosis ~30−60 minutes post release. To sequentially arrest cells in G1 and G2, cells were treated as described for G1 cell cycle arrest. Approximately 8 hours after G1 release, 9 µM RO-3306 was added, and the cells incubated for 16 hours to facilitate G2 arrest. Cells were released as detailed.

### Immunofluorescence

Asynchronous cells were seeded onto pre-sterilised 20 mm glass cover slides placed into each well of a 6-well plate. Any respective drug treatments were carried out as described. To label S-phase cells, 10 µM EdU treatments were included for 20 or 60 minutes prior to cell fixation. To fix cells, the growth media was removed, and the cells washed once in D-PBS (−). Washed cells were fixed using 4% Paraformaldehyde in PBS for 15 minutes at room temperature. Cells were then permeabilised in 0.5% TritonX-100 in PBS for 5 minutes at room temperature and washed twice. If required, EdU or EU Click-IT was carried out as per the manufacturer's protocol (Invitrogen) to detect S-phase cells prior to antibody staining or to measure transcription activity. Antibody staining was subsequently carried out: 100 µl of primary antibody (in

washing buffer: 5% BSA 0.1% Tween-20 in PBS) was added dropwise to cover the glass cover slip and incubated for 2 hours at room temperature. Cells were then washed 3× in washing buffer and 2× in D-PBS (−). Secondary antibody solution was added analogous to primary antibody and incubated for 2 hours at room temperature in the dark. Antibodies used are described: Mouse anti-Ser139 γ-H2AX (Sigma-Aldrich JBW301; 1:1000), Rabbit anti-53BP1 (Novus Biologicals NB100-904; 1:1000), Rabbit anti-S9.6 (Francis Crick Institute; 1:200), Mouse anti-Mitosin (BD Biosciences 610768; 1:300), Rabbit anti-P-Histone H3S10 (Cell Signalling 9701; 1:1000), AF488 anti-mouse secondary (Invitrogen A32723; 1:1000), AF555 anti-rabbit secondary (Invitrogen A21428; 1:1000). Stained cells were washed as detailed before being mounted onto glass slides using DAPI mounting media (Fluoroshield). Slides were dried at room temperature before being imaged using a Leica DM600 Widefield Fluorescent microscope with Leica LASX V. 3.7.4.23463 software. All images were exported as raw grayscale TIFs, analysed using CellProfiler (v4.0.6) and ImageJ (v2.1.0).

## siRNA depletion

HCT116 TRAIP-mAID CMV-OsTIR1 cells were seeded at $1 \times 10^5$ cells/ml and 24 h later, transfected with 50 nM Non-targeting (Horizon Discovery, D-001810-10-05), UBXD7 (Horizon Discovery, L-023533-02-0005) or SPRTN/C1orf124 (Horizon Discovery, L-015442-02-0005) siRNA, using Dharmafect 1 transfection reagent and manufacturer's protocol. Cells were harvested or fixed 72 h later and analysed by western blotting or immunofluorescence. Depletion was confirmed through western blotting, using antibodies: Rabbit anti-UBXD7 (Thermo Fisher Scientific 15779771; 1:1000) and Rabbit anti-SPRTN (Novus Biologicals NBP1-84163; 1:1000).

## Flow cytometry

Three types of flow cytometry experiment were carried out: un-extracted cells, BrdU detection, and extracted cells. For un-extracted cells, following the required experimental procedures (e.g., proliferation curves) cells were harvested and fixed in 70% ethanol in PBS for 16 hours at −20 °C. Following fixation, cells were washed twice in washing buffer (5% BSA, 0.1% Tween-20, PBS) and antibody staining carried out if required. Briefly, washed cells were re-suspended in 100 µl primary antibody in washing buffer and incubated at room temperature for 1 hour, rocking to prevent cells from settling. The cells were washed twice in washing buffer and re-suspended in 100 µl secondary antibody in washing buffer for 1 hour at room temperature in the dark. Stained cells were washed 1× in washing buffer and 2× in D-PBS (−) before being re-suspended in either Hoechst Staining Buffer (5 µg/ml Hoechst 33582, PBS) or Propidium Iodide Staining Buffer (50 µg/ml Propidium Iodide, 50 µg/ml RNase A, PBS). For BrdU detection, 10 µM of BrdU was added to the growth media 1 hour prior to harvesting. Cells were collected and fixed in ethanol as described. Fixed cells were washed once in PBS before being re-suspended in 1 ml 2 M HCL supplemented with 0.1 mg/ml Pepsin for 20 minutes. Cells were then washed, and antibody staining carried out as described. To explore the replisome binding pattern on chromatin, cells were extracted using CSK buffer (25 mM HEPES pH 7.4, 50 mM NaCl, 3 mM MgCl$_2$, 300 mM Sucrose, 0.5% TritonX-100, 1× complete protease inhibitors) to remove soluble fractions. The protocol used to extract cells has been described elsewhere (Forment & Jackson, 2015). Antibodies used are as follows: Rabbit anti-P-Histone H3S10 (Cell Signalling 9701; 1:500); Mouse anti-MCM7 (Santa Cruz 9966; 1:500); Mouse anti-BrdU (BD Biosciences 347580, clone B44; 1:5); AF488 anti-mouse secondary (Invitrogen A32723; 1:1000); AF555 anti-rabbit secondary (Invitrogen A21428; 1:1000); AttoN647 anti-rabbit secondary (Sigma-Aldrich 40839; 1:500). Cells were analysed using a Beckman Cytoflex instrument with CytExpert v2.5 software. Data analysis was carried out using FlowJo (v10.7.1). For all analysis, doublets were first excluded by gating the populations of interest using FSC-A vs SSC-A followed by FSC-A vs FSC-H. Second, gates required for quantification were applied based on control samples, and applied universally to all other samples within the experiment. For example, for cell cycle analysis untreated DNA distributions were empirically gated as G1 (2 N DNA content), G2/M (4 N DNA content), and S-phase (region between 2 N and 4 N). The set gates were then overlaid onto treatment samples to determine any consequential changes. An example of cell cycle stages gating is presented in Supplementary Fig. 2D.

## ChIP-sequencing

For ChIP-sequencing, HCT116 TRAIP degron cells were diluted to $2 \times 10^5$ cells/ml and seeded onto 20 cm dishes (2× per condition). Plated cells were arrested in G1 as described, treated with auxin (IAA) to degrade TRAIP, and released into S-phase. 16 hours after release, cells were harvested and the resulting cell suspension supplemented with 1% Formaldehyde (Sigma-Aldrich) for 10 minutes at room temperature. Formaldehyde crosslinking was quenched using 125 mM Glycine (Sigma-Aldrich). Fixed cells were pelleted (400 g, 3 minutes, 4 °C) and the pellet washed 2× with ice-cold PBS. Cell extraction was then carried out through sequential incubations in ChIP lysis buffer (5 mM HEPES pH 8.0, 85 mM KCL, 0.5% NP40) and ChIP nuclear lysis buffer (50 mM Tris-HCL pH 8.0, 10 mM EDTA pH 8.0, 1% SDS) for 15 and 30 minutes on ice, respectively. Cell lysates were divided into equal aliquots and sonicated (30 amplitude, 15 sec on, 25 sec off, 12 cycles). Sufficient sonication (resulting in 300-500 bp DNA fragments) was confirmed through DNA gel electrophoresis. Chromatin immuno-precipitation was carried out using Protein A Dynabeads conjugated to approximately 1 µg of rabbit anti-Ser139 γ-H2AX (Abcam 29893) or rabbit anti-H2AX (Merck Millipore 07627) antibodies, as detailed in Wang et al.[56]. Immuno-precipitation was validated by quantitative PCR using primers targeting the actin housekeeping gene (Forward: CATGTACGTTGCTATCCAGGC, Reverse: CTCCTTAATGTCACGCAC-GAT). The PCR was performed using AppliedBiosystems Quant-Studio5 with Thermo Fisher Connect software and analysed in Microsoft Excel.

Library preparation was carried out using NEBNext Ultra II DNA Library Preparation Kit for Illumina NEB, as per manufacturer's instruction. The prepared libraries were sequenced using single-end sequencing with a High-75 kit, using Illumina NexSeq instruments. γ-H2AX ChIP-Seq, precision run-on sequencing (PRO-Seq)[33] and RNA Pol II chromatin immuno-precipitation sequencing[34] in HCT116 cells were aligned to the hg38 genome using Bowtie 2 v.2.4.2 on the online platform Galaxy (https://usegalaxy.org[60]). γ-H2AX ChIP-Seq peaks were called in the +IAA treated sample against the γ-H2AX ChIP-Seq in its −IAA control using MACS2 v.2.1.1 with parameters as detailed in[56]. The bedtools intersect intervals function on the online platform Galaxy was used to identify the conserved γ-H2AX ChIP-Seq peaks in the two repeats. The distance between the γ-H2AX ChIP-Seq peak or the TSS with lowest γ-H2AX fold change +IAA/−IAA was calculated using the position of the origins of replication in HCT116 cells provided by Dr Daigaku[31]. The PRO-Seq dataset was used to determine the reciprocal direction between the oncoming replication fork and gene transcription (head-to-head or codirectional), as well as whether the TSS was the first, second, or third and above-transcribed gene encountered. Replication timing for the γ-H2AX ChIP-Seq peak was derived analysing replication timing in HCT116 cells from[30]. The read coverage profiles were generated using the computational environment EaSeq, normalising the γ-H2AX ChIP-Seq file to the H2AX ChIP-Seq file with the function "average"[61].

To identify the list of transcribed genes in HCT116 cells, the counts for each gene were computed by featureCounts on the online platform Galaxy using the annotation of the GENCODE genes (GRCh38.p10) on an ENCODE polyA RNA-Seq hg38 aligned file (ENCFF823JEV). Read per kilobase per million (RPKM) were calculated over each gene and genes that had an RPKM > 1 were considered as transcribed.

The levels of antisense to sense transcription were calculated using the PRO-Seq and RNA Pol II ChIP-Seq datasets using the function "quantify" in EaSeq, for the antisense −1000 bp to the TSS, and for the sense from the TSS to +1000 bp. For the PRO-Seq, as this was strand-specific antisense transcription levels were specifically measured on the strand opposite to the sense transcription. Heatmaps were generated with the function "HeatMap" of EaSeq around TSS ±2500 bp. To identify the genes with the lowest increase in γ-H2AX ChIP-Seq levels following TRAIP depletion, the normalised levels of γ-H2AX ChIP-Seq signal to H2AX was calculated across the TSS ±1000 bp of all transcribed genes in both the repeats and averaged for each gene. Then gene TSS were sorted by the fold change in γ-H2AX ChIP-Seq levels in the +IAA compared to the −IAA.

## ChIP RT-PCR

For the RT-PCR PCR, primers were designed at a series of conserved γH2AX peaks over transcribed genes. ChIPs were performed for γH2AX and H2AX as described above, with γH2AX levels normalised to H2AX levels. γH2AX/H2AX at the target sites were then normalised to the γH2AX/H2AX levels at a negative control region (N2), that is a region in a gene desert on chromosome 13. Cells were grown as above, with triptolide added for 90 minutes. Primers used: BAIAP2 For (CTTTCGTCTCCGTCCTGCTG) and BAIAP2 Rev (GAAGACC CCCAAAGTCCCAG) amplifying 278 bp product; RAC3 For (TGTGATAC ATTCTGGCCCCG) and RAC3 Rev (GAACCCCCAGACGGACAG) amplifying a 179 bp product; P4HB For (CGGATTGGACACTCACACCA) and P4HB Rev (CAGAGTCCGTGCTACCGAAA) amplifying a 223 bp product; WDR45B For (CACCGTGGTCCTGGTTGAAG) and WDR45B Rev (CATG AACCTCCTGCCGTGTA) amplifying a 71 bp product; N2 For (AGCT ATCTGTCGAGCAGC) and N2 Rev (CATTCCCCTCTGTTAGTGGAAGG) amplifying a 112 bp product.

## Proximity ligation assays

PLA were carried out using the DuoLink PLA Kit. The provided manufacturer's instruction was optimised for use on cell suspensions. Parental cell lines were diluted to $2 \times 10^5$ cells/ml and seeded to 60-mm tissue culture dishes. Cells were arrested in G1, where auxin (IAA) was added, and released into S-phase. Approximately 12 hours following cell cycle release, 10 μM EdU was added for 20 minutes to label nascent DNA. For PLA reactions using antibody recognition only, cells were harvested approximately 12.5 hours following cell cycle release to provide a similar timepoint to that when incorporating EdU treatment. Cells were then harvested and extracted using CSK buffer as described previously. Extracted, permeabilised cells were subjected to the Click-It reaction to conjugate biotin to incorporated EdU (Invitrogen, as per manufacturer's protocol). Cells were washed 2× in PLA washing buffer and re-suspended in 50 μl of each primary antibody made up in washing buffer (elongating RNA Polymerase II (serine 5): Mouse anti-Rpb1 CTD (Cell Signalling 2629 S, 4H8; 1:250); Rabbit anti-biotin (Bethyl Laboratories A150-109A; 1:500); Rabbit anti-GFP (Chromotek PABG1; 1:250); Rabbit anti-AND 1 (Novus Biological NBP1-89091; 1:250); Mouse anti-GFP (Roche 11814460001; 1:250)) overnight at 4 °C. Following primary antibody incubation, cells were washed 3× in PLA wash buffer A and incubated in 30 μl secondary antibody solution (6 μl '+', 6 μl '−', 18 μl 3% FBS in PBS) for 100 minutes at 37 °C. Cells were washed again 3× in washing buffer A and re-suspended in 30 μl Ligation Buffer, incubated for 60 minutes at 37 °C. Any interacting PLA probes were then amplified using a rolling circle assay. Washed cell pellets were re-suspended in PLA Amplification buffer for 100 minutes at 37 °C. Finally, cells were washed 2× in washing buffer B, re-suspended in 0.01× washing buffer B diluted in distilled $H_2O$. Diluted cell suspensions were added to 20 mm cover slips and centrifuged to adhere cells to cover slips (400 g, 5 min). Slides were mounted using DuoLink PLA DAPI mounting media. Cells were analysed using a Leica DM600 widefield microscope and analysed using CellProfiler. For analysis,

nuclei regions of interest were first segmented using the DAPI signal. Only PLA signal residing within the nuclei were quantified.

## Statistical analyses

All statistical analyses, except from ChIP-Sequencing data, was carried out using RStudio (v 1.0.153). The imported data was first subject to normality testing through qqplots to determine the appropriate statistical testing method. All plots were created using ggplot2 and the plug-in ggpubr[62]. Two biological repeats for each γ−H2AX and H2AX ChIP-Seq in −IAA and +IAA were analysed, with repeats assessed for correlation before being combined. Student $t$ test and Mann–Whitney $t$ test were calculated using the software Prism (GraphPad). All statistical analyses were two-sided unless otherwise stated in the figure legend. All measurements were taken from distinct samples and no repeated measurements were taken.

## *X. Laevis* methods

All the work carried out with *X. Laevis* egg extract was approved by the University of Birmingham Ethics Committee and by UK home office project license no: P081C27D8.

## Inhibitors and recombinant proteins

EcoR1 (R6011, Promega) was purchased at stock 12 U/μl and added to the extract at 0.05 U/μl, Aphidicolin (A0781, Sigma) was dissolved in DMSO at 8 mM and added to the extract along with demembranated sperm nuclei at 40 μM. Caffeine (C8960, Sigma) was dissolved in water at 100 mM and added to the extract along with demembranated sperm nuclei at 5 mM. MLN4924 (A01139, Active Biochem) was dissolved in DMSO at 20 mM and added to the extract 15 minutes after addition of sperm nuclei at 10 μM.

Recombinant His-tagged *X. Laevis* CyclinA1 NΔ56 (pET23a-*X.l* cyclinA1 NΔ56), was expressed and purified as previously described[21]. CyclinA1 NΔ56 was used at a final concentration of 826 nM in the egg extract to drive the extract into mitosis.

## Antibodies

Mouse anti-PCNA (Sigma P8825; 1:2000); Rabbit anti-TRAIP (Novus Biologicals NBP1-87125; 1:500); Rabbit anti-P-Chk1 (S345) (Cell Signalling 2341; 1:1000); Rabbit anti-Ser139 γ-H2AX (Trevigen 4418-APC-020; 1:1000).

Affinity-purified anti-Cdc45, anti-Psf2[63], anti-Mcm7[21], and anti-GINS antibody[64] were previously described. Affinity-purified anti-TRAIP is described in Supplementary Fig. 12.

Recombinant *X. Laevis* SUMO-TRAIP was purified as previously described[21] and was also used for raising antibodies in rabbits. The resulting antibody sera was purified in-house against the purified antigen. The specificity of this new antibody is presented in Supplementary Fig. 12A.

## *X. Laevis* egg extract preparation

Metaphase II arrested egg extracts were prepared as previously described[65] from unfertilised female frog oocytes. In order to increase the amount of stage 6 (mature) oocytes, 10 frogs were primed with 150 units follicle stimulating hormone Foligon (Intervet) 2–7 days before eggs were required. Frogs were injected with 400–600 units of serum gonadotropin Chorulon (Intervet), and subsequently transferred to laying tanks containing 2.5 l of 1× MMR (0.1 M NaCl, 2 mM KCl, 1 mM $MgCl_2$, 2 mM $CaCl_2$, 0.1 mM EDTA, 5 mM HEPES, pH to 7.8 with NaOH). Frogs were kept in the tanks laying eggs overnight at ≤ 23 °C. Eggs from different frogs were collected the next morning in a 1 l glass beaker.

Good quality eggs were rinsed with 1× MMR and de-jellied by rinsing them in cysteine solution (2.2% cysteine, 5 mM EGTA, pH to 7.6 with KOH). De-jellied eggs were rinsed again with 1× MMR, and then washed in UEB buffer (50 mM KCl, 50 mM HEPES, 5 mM $MgCl_2$, 5 mM EGTA, 2 mM DTT, pH to 7.6 with KOH) and white/swollen apoptotic

eggs floating on the top were removed. The de-jellied eggs were packed into 14 ml round bottom polypropylene tubes (187261; Greiner) with 1 ml UEB containing 10 µg/ml protease inhibitors: aprotinin, leupeptin, and pepstatin and 50 µg/ml Cytochalasin D (C8273-5MG, Sigma). The tubes were then spun to pack the eggs in a Beckman JS13 rotor at 800 × g for 1 minute, room temperature (RT). The white apoptotic swollen eggs that float to the top were removed followed by another centrifugation at high speed at 10,000 × g for 10 min at RT. This results in separating the eggs into a lipid layer at the top, brown cytoplasmic fraction in the middle, and an insoluble egg yolk pellet at the bottom. The cytoplasmic layer was collected using a 20 G needle and a 1 ml syringe via side puncture. Extract was supplemented with 10 µg/ml protease inhibitors, 10 µg/ml Cytochalasin D, and 15% of LFB1/50 (10% sucrose, 50 mM KCl, 40 mM HEPES pH 8, 20 mM K phosphate pH 8, 2 mM MgCl$_2$, 1 mM EGTA, 2 mM DTT, 1 µg/ml of each: aprotinin, leupeptin and pepstatin). The extract transferred to SW55 ultracentrifuge tubes (344058, Beckmann) which were then subjected to a final clarifying spin at 30,000 × g for 20 min at 4 °C. After the spinning, the yellow lipid plug from the top was removed, and the pale yellow cytoplasmic fraction collected. The collected extract was then supplemented with 1% v-v of glycerol, and frozen in liquid nitrogen in small beads and stored at −80 °C.

### DNA synthesis assay
Interphase *X. Laevis* egg extract was supplemented with 10 ng/µl of demembranated *Xenopus* sperm nuclei and incubated at 23 °C for indicated times. Synthesis of nascent DNA was then measured by quantification of radiolabelled α$^{32}$P-dATP (NEG512H250UC, Perkin Elmer) incorporation into newly synthesised DNA, as described before[65].

### Chromatin isolation time-course
Interphase *X. Laevis* egg extract was supplemented with 10 ng/µl of demembranated sperm DNA and subjected to indicated treatments. The reaction was incubated at 23 °C for indicated length of time when chromatin was isolated in ANIB100 buffer (50 mM HEPES pH 7.6, 100 mM KOAc, 10 mM MgOAc, 2.5 mM Mg-ATP, 0.5 mM spermidine, 0.3 mM spermine, 1 µg/ml of each aprotinin, leupeptin and pepstatin, 25 mM β-glycerophosphate, 0.2 µM microcystin-LR and 10 mM 2-chloroacetamide (Merck) as described previously[65]. To study mitosis, the interphase extract was supplemented with MLN4924 and allowed replication to complete. Once completed extract was treated with CyclinA1 NΔ56.

During the chromatin isolation procedure, a sample without addition of sperm DNA (no DNA) is processed in an analogous way, usually at the end of the time course, to serve as a chromatin specificity control. The bottom of the PAGE gel on which the chromatin samples were resolved was cut off and stained with Colloidal Coomassie (SimplyBlue, Life Technologies) to stain histones which provide loading controls and indications of sample contamination with egg extract (cytoplasm).

### Nuclei isolation for Chk1 phosphorylation
The nuclei isolation was performed as previously described[66].

### Immunodepletion
TRAIP immunodepletions were performed using Dynabeads Protein A (10002D, Life Technologies) coupled to *Xenopus* TRAIP antibodies raised in rabbits and affinity purified or non-specific rabbit IgG (I5006, Sigma). The TRAIP antibodies were coupled at 600 µg per 1 ml of beads. Effective immunodepletion required 2 rounds of 1 h incubation of egg extract with antibody-coupled beads at 50% beads ratio (e.g., 2 rounds of 100 µl of egg extract incubated with 50 µl of coupled beads).

### Reporting summary
Further information on research design is available in the Nature Portfolio Reporting Summary linked to this article.

## Data availability
The sequencing data discussed in this publication have been deposited in NCBI's Gene Expression Omnibus and are accessible through GEO Series accession number GSE201158. Source data are provided in this paper.

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

## Acknowledgements
This work was supported by the BBSRC-funded MIBTP studentship, JSPS Summer programme, and BBSRC funded MIBTP Career Development Fellowship for S.S. Wellcome Trust Investigator Award (215510/Z/19/Z) funded R.J and A.G, while BBSRC BB/T001860/1 funded A.R.-W. The University of Birmingham, BBSRC (BB/S016155/1) and Cancer Research UK (C17422/A25154) to M.S.

## Author contributions
S.S., R.J., M.M.S., D.P., A.F., A.R.-W., and C.F.C. acquired and analysed the data within the manuscript. P.R. and M.S. analysed the ChIP-seq data. T.N. and M.T.K. assisted with the generation of TRAIP-mAID and TRAIP-mAC HCT116 and RPE cell lines. M.S. and A.G. designed the study and wrote the manuscript.

## Competing interests
The authors declare no competing interests.
