## [Peer Review File · Nature Communications]

Ubiquitin ligase TRAIIP plays an essential role during the S-phase of unperturbed cell cycle in the resolution of DNA replication – transcription conflictsREVIEWER COMMENTS

Reviewer #1 (Remarks to the Author):

This manuscript investigates the impact of TRAIP loss on cell cycle progression and DNA damage accumulation. It was previously reported that TRAIP may have multiple roles in cell cycle progression and genome stability, but TRAIP studies are generally difficult because TRAIP is essential for cell proliferation and thus its genetic ablation causes proliferation arrest.

To overcome this, the authors employed an inducible degron approach, which is introduced by CRISPR-mediated genomic manipulation at the endogenous TRAIP locus in colon carcinoma HCT116 cells. This results in fast TRAIP degradation in a controllable manner, allowing the authors to investigate the impact of TRAIP loss in each cell cycle stage.

The authors find that TRAIP is essential in S-phase: in its absence, cells accumulate in G2 in a Wee1-dependent manner and undergo senescence. Separately, by performing gH2AX ChIP-Seq, they show that DNA damage accumulates at transcription start sites (TSS) in TRAIP-depleted cells, which could be suppressed by inhibiting RNA PolIII-mediated transcription. Finally, using proximity ligation assays they show an increased co-localization between PolII and nascent DNA in TRAIP-depleted cells. The authors conclude that TRAIP could help resolve replication-transcription conflicts.

The topic is important and timely, and some of the results are intriguing. Some aspects of the manuscript are technically sound (particularly the use of multiple clones) and the ChIP-seq experiments are powerful and informative. However, the manuscript presents few mechanistic insights. Most of the work is simply validating previous studies performed with genetic inactivation of TRAIP. In addition, there is no clear demonstration of the proposed model (that TRAIP resolves replication-transcription conflicts) but rather this is indirectly implied. Finally, various parts are strenuously connected to each other, making the relevance of these different aspects unclear. As such, I believe that in its current form the manuscript is not suitable for publication in Nature Communications.

Specific comments:

1. The authors do not convincingly demonstrate that TRAIP suppresses replication-transcription collisions, as implied many times throughout the manuscript including in the title. The PLA in fig5C only shows PolIII recruitment to nascent DNA, but this could potentially simply reflect transcription of newly-replicated DNA, rather than replication-transcription collisions. The authors should perform PLA experiments measuring PolIII interaction with replication forks components such as PCNA or PolDelta. Moreover, it may be informative to measure R-loops, or other outcomes of transcription-replication collision.

2. There is only limited mechanistic insight presented. The authors mostly speculate on the possible mechanism. For the manuscript to be suitable for Nature Communications, I believe that uncovering (some of) the mechanism is essential. The authors need to show that TRAIP ubiquitinates PolIII and this results in its removal by p97. The possible involvement of Spartan, and how this may be connected to TRAIP, needs to be investigated as well.

3. The gH2AX ChIP-seq results are interesting, but in their current form do not inform much mechanistically. The authors should extend the analyses to investigate if a similar impact on gH2AX accumulation at TSS occurs upon depletion of other proposed components of the pathway, such as p97 and Spartan. Moreover, they need to test if this can be suppressed by inhibiting transcription (see comment below)

4. An important caveat is that the observed phenotypes could be cell line specific. All experiments were done in a single cell line. The authors need to expand the analyses to other cell lines.

5. Can the authors rule out that TRAIP is required for the G1/S transition? I am not convinced that this can be ruled out based on the experiments presented in Figure 3.

6. Showing the percentage of cells with foci (Fig5) is not the appropriate and established way to depict this quantification. The authors should present dot plots of the number of foci/cell, which show the distribution of the data, and may change the data interpretation.

7. Simply measuring foci formation (Fig5) is not convincingly demonstrating that the damage observed is transcription-dependent. Investigating the impact of transcription inhibitors in gH2AX CHIP-seq experiments such as shown in Fig4 is a much more powerful approach.

8. Figure 5C: This experiment does not demonstrate that TRAIP actively suppresses PolII collision with the replication machinery. For example, can the authors rule out that the increase in EDU-PolII PLA is instead caused by increased recruitment of PolII on nascent DNA upon TRAIP depletion?

9. The relevance of Wee1 is unclear. The authors show some data early in the manuscript, but how the findings downstream (gH2AX accumulation at TSS, PolII engagement etc) relate to Wee1-mediated checkpoint is unclear, and needs to be addressed.

10. A similar comment regarding senescence: Why does increased polII-mediated DNA damage lead to senescence (and not, for example, apoptosis)? Can transcription inhibition suppress senescence induced by TRAIP depletion?

Reviewer #2 (Remarks to the Author):

During the last few years, TRAIP has been reported by several groups to be essential for cell viability, embryonic development, and overall to play broad roles in the maintenance of genome integrity. TRAIP has been characterized in many processes including the repair of DNA lesions occurring upon various genotoxic agents. It is a replisome-associated ubiquitin ligase, mutated in the microcephalic primordial dwarfism, which helps the replication machinery to overcome impediments, such as interstrand crosslinks and protein-DNA crosslinks, which threaten genome stability.

Despite already known function of TRAIP in DNA repair processes, in the present manuscript, Scaramuzza and colleagues aimed at understanding which of the numerous functions of the E3 ubiquitin ligase TRAIP is the most crucial for cell viability, cell proliferation and prevention of microcephalic dwarfism. The authors developed a degran-

based system to study the effect of loss of TRAIP in specific phases of the cell cycle. As already published, authors confirmed that degradation of TRAIP causes a G2 arrest of the cell cycle and cell senescence. They also showed that TRAIP plays essential role in the S phase and is required to resolve replication-transcription conflicts. Indeed, its depletion results in the accumulation of DNA lesions at the transcription start sites, as also shown by γ H2AX CHIP-Seq analysis.

Overall, the paper is interesting, very well organized and increases current knowledge on a key factor regulating genome integrity at multiple levels.

There are some major concerns that dampen enthusiasm at this stage and should be addressed. These relate mainly to the limited cellular systems used and the lack of mechanistic insights. I will explain these in more detail below.

In addition, there are some minor suggestions that would help support their findings.

Major comments:

1. The first major concern regards the use of a single cell system, the HCT116 colorectal cancer cell line, for all the functional data, raising the issue of whether the observed phenomenon is restricted to the specific cell line type. Also, the data obtained using the TRAIP depletion in cell-free *Xenopus* egg extracts (Figure 6) are indirect (absence of evidence is not an evidence of absence). Authors should recapitulate the key data using a different cell system.

2. The second major point that needs better investigation is the mechanistic understanding of TRAIP's role in resolving replication-transcription conflicts. Is this function rescued by the re-expression of TRAIP in the AID system they have developed? Does it depend on its ubiquitin ligase activity? The use of a RING finger mutant of TRAIP will easily help address this point. If the RING finger is required, what are the targets involved in this function?

3. Figure 2D: in the paper authors show that damage occurs in the S phase upon TRAIP depletion. Since 24h after IAA induction cells already accumulate in G2 (as shown in Figure

1D), it is likely that the resulting foci count (for γ H2AX and 53BP1) refers to that of cells arrested in G2. To clarify this point, the authors should do the same IF analysis upon EdU labelling (since a similar approach has already been used in Supplementary figure 6A) and/or G2 specific markers.

4. Figure 3A: authors must include earlier time points (and also additional information stating the proper S phase staging, e.g. PCNA profile) in order to show whether the S phase progression is unaffected upon TRAIP depletion. The observed damage accumulation observed in Figure 2D might be mainly due to the faster replication rate.

5. Figure 8: authors showed that ATR inhibition in TRAIP depleted cells causes elevated DNA damage but can rescue G2 arrest (Supplementary figure 5A). In the last sentence authors propose the combinatorial treatment of ATRi and TRAIP depletion, which appears a bit contradicting with their own findings. Please explain and discuss better this concept.

6. Supplementary figure 5A: Authors showed that ATRi but not ATMi can rescue the G2 cell cycle arrest upon TRAIP depletion, proposing that damage originating from S phase. Authors should show the activation of the canonical ATR substrate upon TRAIP depletion (Western blot showing pRPA, pCHK1 etc...).

Minor comments:

1. Figure 1B: colour coding for the graph is missing.
2. Figure 1C: Y axis should be better explained. The fold change after 24 hours of +/- IAA already shows almost 5-fold change.
3. Figure 1D: for DNA content, authors need to mention if specific model system is applied (example: Watson pragmatic model)?
4. Figure 2A: scale bars disturb the IF images. Show only bar (better 10 μ m) and mention the dimension in the legend.
5. Figure 3: to substantiate the finding that DNA damage originates from S phase upon TRAIP depletion, authors should monitor γ H2AX.
6. Figure 4A: present the representative image showing γ H2AX hotspot with TSS

bidirectionality.

7. Supplementary figure 9B: authors used RNase A treatment as control for S9.6 antibody. Since RNase A is known to degrade free RNA, authors should report the exact experimental conditions used for the assay. Authors should also include RNase H treatment as control.

Reviewer #3 (Remarks to the Author):

This paper explores the role of the essential protein TRAIP in DNA replication and the maintenance of genome integrity using acute depletion by auxin-inducible degradation. The authors have developed an elegant and robust method to deplete untagged or mClover-tagged TRAIP in HCT116 cell lines. Using these cell lines combined with cell cycle inhibitors, the authors find that TRAIP likely acts during S phase to suppress DNA damage and the absence of TRAIP triggers a Wee1-dependent G2 arrest. Further experiments investigate the role of TRAIP in S phase, suggesting a role for TRAIP in suppressing transcription-associated replication stress.

The authors then switch to *Xenopus laevis* egg extracts to determine the impact of TRAIP on replication in the absence of transcription. While compelling, again this referee found the data incompletely described making it difficult to follow experiments. Like earlier figures, controls such as caffeine and aphidicolin were not described in the text at all, rather the method, rationale and sometimes their conclusions were described in figure legends. Since many of the experiments contain multiple steps, walking through the method and rationale makes sense in many cases but adding conclusions/interpretations is not appropriate for a figure legend. Titling an entire figure with a conclusion is appropriate but doing so for each panel is not (Figure 3). Further, the lack of agreement between published literature, *Xenopus laevis* extracts and HCT116 cells for mitotic experiments was only briefly stated and then not discussed in either the results or discussion. Together, this made it difficult to interpret these experiments.

Possibly due to the sheer volume of data, the description of the results and author interpretations/conclusions were difficult to follow and at times convoluted. These need to be separated to facilitate readers drawing their own conclusions from the data presented.

Other issues such as not explicitly defining abbreviations in the text or even in the figure legends (see points 7-Cod,Hth and 16-CULi) put the onus on the reader to make assumptions to follow the manuscript.

While the experimental system and initial figures were presented elegantly, the manuscript became disorganized later on; multiple supplementary figure panels were not referred to at all, details missing from other figure panels, and referencing figures out of order. Further, placing p values in long lists at the end of a figure legend (sometimes 8 p values in a list for one panel) means the reader must do the work rather than having the figure speak for itself. It is difficult to follow the argument of a manuscript when an entire figure with three panels (24 images with 2 drug treatments and 4 separate graphs with 6 bars each) is introduced in the second-to-last sentence of the discussion. This means many data panels are not properly presented in the results.

Major comments

1) Line 64-65: “Genome-wide, such unreplicated regions correlate with common fragile sites (CFS), which are chromosomal loci responsible for the majority of rearrangements found in human disease”

- This sentence is incorrect due to citation of a review that does not accurately reflect the literature they cite. The review refers to Burrow et al 2009 for their overly broad statement “Strikingly, recent cancer genome-sequencing projects have revealed that more than half of the DNA rearrangements in cancers cluster within certain chromosomal loci known as common fragile sites (CFSs)...” which is not accurate. Rather Burrow et al 2009 uses fragile sites as a term for both common and rare – 52% of translocation breakpoints mapped to fragile sites -- thus this sentence is incorrect on 3 levels. 1) they only looked at translocations with gene pairs, 2) they compared to common + rare fragile sites and 3) this is only for human cancers not human disease. Delete or rewrite citing the primary literature and what was found.

2) The pan-nuclear H3S10-phospho staining looks strange. Mitotic H3S10 phosphorylation normally starts in discrete foci and then signal “spreads” as chromosomes condense. This means the signal usually looks like worms or squiggles, or other textures associated with condensing chromosomes. The signal here looks like DAPI.

The DAPI staining also does not reflect a traditional mitotic cell with condensed or condensing DNA, and rather looks like an interphase cell. CENPF also marks G2 cells.

3) Line 534-536 – This sentence is confusing: “Whilst overall lower proportions of cells were detected incorporating EdU/BrdU, no differences in the average amount incorporated per positive cell were found (Supplementary Figure 6A,B).”

Supplementary Figure 6A: What is being measured in the graphs?

- The Y-axis in Supplementary Figure 6A graph not clear, is it missing a word? Mean cells positive for EdU \diamond is this the % of cells (because this is FACS analysis I assume it is this), or # of cells? Or is it mean BrdU intensity? The red does appear shifted down in -/+IAA overlay but this would be intensity which authors state is unchanged.... The graphs vs data are confusing— if S6A graph is the % of cells, show the gates on the data with #s like done for H3S10P in Supplementary figure 3A. How the graphs in Supplementary figure 6B reflect the primary data from the images is much clearer – the Y axis in top graph clearly states what is being measured.

4) Line 466: “... G2 cell cycle arrest mediated by WEE1.”

\diamond Change to “dependent on Wee1”? The arrest may be initiated/mediated by other components but the G2 arrest is maintained Wee1 so indeed it is required... and agree inhibition reduces G2 levels down to the -IAA control.

5) Figure S3 – please show representative image of condensed chromosomes to pair with S3B

S3D - is this the same antibody used in Figure 2A (mitosin) and now using cenpF? Be consistent in naming scheme. Also there is no mention of S3D in the text.

6) Figure S3 – When did you add Wee1, after 24h of IAA or in the last 1/3 hours of a 24h

treatment? Can you make a scheme?

7) Line 489-91: "Moreover, inhibition of ATR, but not ATM, was able to rescue the observed G2 cell cycle arrest (Supplementary Figure 5A) suggesting that the damage is created in S-phase during DNA replication."

This is not necessarily true, as ATR is also involved in G2/M checkpoint activation & inhibition of ATR activity can also abrogate replication-independent G2/M checkpoint initiated by UV (Stiff et al 2008).

8) 547-549 – Conclusions and results description here are tangled and should be separated. It would help to state results independently, then state interpretation.

9) Supplementary 7A – it is not explained at all why they use APH and what they are comparing.

Please describe.

10) Include the name of the gene and the chromosomal coordinates shown in figure 4A – the data will become public therefore readers should be able to assess the loci presented with available data.

11) Lines 588-591: Data presented in 4F and in the text is confusing and does not match. The numbers 75% for hotspots and 65% for low is not much of a difference but this is not even reflected in the graph. We must assume Cod stands for codirectional... Hth for head-to-head... Please define all acronyms in the figure legend text.

◇ Do the #s reflect Cod1 + Hth1? If so, you must state this clearly.

* There are no P values stated in the text or figure legend, is this significant? If it is not, please state.

12) When the authors talk about antisense transcription are they talking about actual antisense transcription where the opposite strand of the annotated gene is being partly transcribed or is it bidirectional transcription like they show in Fig 4A? Can they show

browser tracks that show the antisense portion? Many promoters are actually bidirectional, with genes on both the - and + strands; these regions may be enriched there.

It is not clear if the signal from PROseq is antisense transcription in Fig 4A. It may be an unannotated (possibly abortive) transcript going the opposite direction of the annotated gene. Without clear information on the scale of the regions shown (it seems the tick marks are 1 kb apart but the font is so tiny), it is difficult to gauge.

13) Maintain consistency between figure panels:

- Figure 5A % of cells positive for 53bp1 vs. 5B % of cells with 3 > foci – please sure same measurements for both. If they are the same, please use the same Y axes for both

14) Figure 5B – the graphs for 53BP1 do not match shown pictures -

– this may be problem of intensity but this should be described – something is off.

15) I do not think you can say DRB and TRAIP act in same pathway because DNA damage is not additive since the DNA damage is being induced in S phase. The #s for DRB alone are already 50% of cells... It that the majority of cells without DNA damage are in G1?

16) Figure 5C – pictures need to be scaled the same with scale bars in same place.

- need to count foci only in nucleus (DAPI-overlapping), not just cell... there are dots outside nucleus as well. Was this done? If so, please state.

The pictures are fuzzy and not in focus. Show higher resolution images?

17) Line 698 (and throughout) - While the authors suggest TRAIP's role in S phase is the primary role for viability, this is not strictly shown as they did not actually perform viability experiments with cells depleted for TRAIP only in G2/M. They do show that its role in S phase is responsible for triggering the G2 arrest, therefore this referee agrees that the S phase role is very likely why TRAIP is essential. Thus, author conclusions should be softened from "demonstrate" or "we find" to "strongly indicate" or "we conclude".

18) Line 707-708 – replication-transcription collisions were not measured – sites of co-

occurring replication and transcription were identified – potential interactions were increased.

19) Line 639-641 / discussion: TRAP may not be involved in the resolution of conflicts – this has not been demonstrated. Another possibility is that it prevents transcription-replication collisions from occurring in the first place. Prevention would lead to a similar outcome as promoting repair. This (and other?) alternate possibility should be discussed. Either way it is interesting.

20) Refer to the figures in order and describe all results in the results section — supplementary figure 9b is in the discussion but never mentioned in the text.

21) Supplementary figure 9 uses inappropriate control. RNase A digests ssRNA and to a lesser extent RNA:DNA hybrids – not sure why or how this is used as a control. The S9.6 antibody when used for IF can recognize other RNAs as well as RNA:DNA hybrids (see Smolka et al 2021). More appropriate controls would be RNase H to specifically digest hybrids and RNase T1-resistant signal? A dot blot, slot blot or gel electrophoresis a better way to measure bulk RNA:DNA hybrids.

22) The S9.6 staining looks different from other published versions – including HCT116 cells. Normally S9.6 clearly lights up the nucleoli – here the staining is faint and throughout nucleus (as well as in cytoplasm, which is normal).

23) Figure 8 is after the model and only mentioned in the last sentence of the discussion! What is happening with the organization?

Minor comments

1) Figure 1b – needs legend

2) Need to cite Addgene plasmids in the manner dictated by the website – “kind gift from ...”

and often the paper used.

3) Supplementary 5C: move p-value lines over the appropriate bars in graph, they seem to have gotten shifted out of place. Check right pane

4) Line 704-706 – a lot of commas. The complex sentence structure makes it hard to follow the line of thought.

5) Continuous repetition of “auxin (IAA)”. I recommend altering the first instance to auxin (+IAA). Once +IAA is defined, it is no necessary to put auxin every single time in parentheses.

6) Define abbreviations – CULi – link to description in the text

7) Figure 6C – add explanation of APH and APH + caff in main text as positive control? This was a helpful control but only described in legend. Label blot properly for Chk1 as phospho-chk1.

8) Figure 2B, Figure 3, and others: Having over 5 p values listed at the end of figure descriptions in legend makes it slow to match each p value to each condition. Either place the precise p value over the bar it represents instead of the asterisks (especially since the asterisks are not defined in the figure legend) OR define the p value asterisks as > 0.05 or > 0.005 , etc.... Either one would make it easier to decipher.

9) Mention that 2 independent clones were examined throughout in main text at least once? Only stated in figure legends.

Point by point response to reviewers' comments.

Reviewer #1 (Remarks to the Author):

We are happy to see that the reviewer thinks that:

The topic is important and timely, and some of the results are intriguing. Some aspects of the manuscript are technically sound (particularly the use of multiple clones) and the ChIP-seq experiments are powerful and informative.

The reviewer has a number of specific comments that we addressed.

Specific comments:

1. The authors do not convincingly demonstrate that TRAIP suppresses replication-transcription collisions, as implied many times throughout the manuscript including in the title. The PLA in fig5C only shows PolII recruitment to nascent DNA, but this could potentially simply reflect transcription of newly-replicated DNA, rather than replication-transcription collisions. The authors should perform PLA experiments measuring PolII interaction with replication forks components such as PCNA or PolDelta. Moreover, it may be informative to measure R-loops, or other outcomes of transcription-replication collision.

We have taken the reviewer's comments on board and performed PLA between RNA PolII and replication forks components PCNA and AND-1 (Figure 5E). We have also measured levels of R-loops upon degradation of TRAIP using R-loop specific S9.6 antibody (Supplementary Figure 11D).

2. There is only limited mechanistic insight presented. The authors mostly speculate on the possible mechanism. For the manuscript to be suitable for Nature Communications, I believe that uncovering (some of) the mechanism is essential. The authors need to show that TRAIP ubiquitinates PolII and this results in its removal by p97. The possible involvement of Spartan, and how this may be connected to TRAIP, needs to be investigated as well.

In response to this comment, we are now showing that recombinant TRAIP can ubiquitylate RNA PolII *in vitro* (Figure 7B), *in vivo* we find TRAIP in proximity of RNA Pol II in S-phase (Figure 7A), while the level of ubiquitylated RNA PolII on chromatin during S-phase is reduced upon TRAIP-mAID degradation (Figure 7D) suggesting that TRAIP is likely to ubiquitylate RNA PolII.

We and others have shown elsewhere that treatment of cells with p97 inhibitors stops their ability to progress through S-phase due to creation of DNA damage. We have avoided therefore using these inhibitors due to multiple roles of p97 during this, and other, cell cycle stages. We have, however, investigated which of the p97 cofactors can collaborate with p97 and TRAIP in this process and shown that downregulation of UBXD7, but not Spartan, is epistatic with TRAIP-mAID degradation in terms of DNA damage creation during S-phase (Figure 7D).

3. The γ H2AX ChIP-seq results are interesting, but in their current form do not inform much mechanistically. The authors should extend the analyses to investigate if a similar impact on γ H2AX accumulation at TSS occurs upon depletion of other proposed components of the pathway, such as p97 and Spartan. Moreover, they need to test if this can be suppressed by inhibiting transcription (see comment below)

γ H2AX ChIP-seq analyses are time and material consuming and expensive to use them to validate different steps of this mechanism. We have therefore selected 4 TSS with bidirectional transcription for which we could design primers able to efficiently and cleanly amplify the TSS region using RT PCR. Using γ H2AX ChIP RT-PCR method, we have validated that inhibition of transcription with triptolide indeed decreased level of γ H2AX signal at these TSS.

As for p97 and Spartan – please see comment to point 2.

4. An important caveat is that the observed phenotypes could be cell line specific. All experiments were done in a single cell line. The authors need to expand the analyses to other cell lines. Technically, we have used 4 different cell lines – 2 clones of TRAIP-mAID and 2 clones of TRAIP-mAC throughout the described work. However, we appreciate that they all have the same genetic background of parental HCT116 cell line. We created therefore a TRAIP-mAID degron in immortalised but not transformed retinal pigment epithelium RPE1-hTERT cell line. We considered other cell lines but tried to avoid HeLa cell line due to their chromosomal instability and variable p53 levels; while commonly used in DNA replication studies U2OS cells are triploid, making CRISPR modifications more tricky.

RPE1 cells are a good control for conservation of this process in non-cancerous cells, but are large and grow slowly with reduced proportions of cells in S-phase, making cell cycle analyses very challenging (Villa, Fujisawa et al. 2021). Nevertheless, we have confirmed the main phenotypes observed upon TRAIP-mAID degradation in this cell line: decrease in cell viability (Figure 1B), cell proliferation (Supp Figure 2C), accumulation of DNA damage foci (γ H2AX and 53BP1) upon TRAIP-mAID degradation (Supp Figure 5A), accumulation of DNA damage specifically in S-phase cells (Supp Figure 8C) and dependence of this damage generation in S-phase on presence of RNA PolIII on DNA (Figure 5B).

5. Can the authors rule out that TRAIP is required for the G1/S transition? I am not convinced that this can be ruled out based on the experiments presented in Figure 3.

None of our cell cycle experiments upon TRAIP-mAID or TRAIP-mAC degradation ever shown any accumulation of cells in G1 stage of the cell cycle (Figure 1 and Figure 3), suggesting that the progression from G1 to S phase is normal. We have, however, performed the suggested experiment – we degraded TRAIP at the end of G1 arrest, released cells from lovastatin arrest and monitored their cell cycle progression into S-phase at earlier time points than previously. These data are now presented in Supp Figure 6A and show no differences between -/+ IAA samples.

6. Showing the percentage of cells with foci (Fig5) is not the appropriate and established way to depict this quantification. The authors should present dot plots of the number of foci/cell, which show the distribution of the data, and may change the data interpretation.

Due to the undamaged nature of our experiments, the differences observed were often subtle, and so showing the % of cells appeared appropriate. However, we take the comment onboard and we are now showing both types of the graphs in Figure 5 and Supp Figure 10B.

7. Simply measuring foci formation (Fig5) is not convincingly demonstrating that the damage observed is transcription-dependent. Investigating the impact of transcription inhibitors in γ H2AX ChIP-seq experiments such as shown in Fig4 is a much more powerful approach.

As explained in point 3 we have now utilised γ H2AX ChIP RT-PCR to monitor level of γ H2AX signal at selected TSS +/- triptolide.

8. Figure 5C: This experiment does not demonstrate that TRAIP actively suppresses PolIII collision with the replication machinery. For example, can the authors rule out that the increase in EDU-PolIII PLA is instead caused by increased recruitment of PolIII on nascent DNA upon TRAIP depletion?

To answer this point we have checked that there is no change of chromatin level of RNA PolIII upon TRAIP degradation (Supp Figure 11B) and that we observe no increase in general level of transcription in cells by monitoring level of synthesised RNA (Supp 11A). We have also confirmed our finding of increased proximity of RNA PolIII and replisomes through PLA between RNA PolIII and PCNA and AND-1 (Figure 5E).

9. The relevance of Wee1 is unclear. The authors show some data early in the manuscript, but how

the findings downstream (gH2AX accumulation at TSS, PolII engagement etc) relate to Wee1-mediated checkpoint is unclear, and needs to be addressed.

WEE1 is used by us as a gatekeeper of G2/M checkpoint. WEE1 is the kinase that ultimately stops cells experiencing replication stress and DNA damage from progression into mitosis. Our use of this inhibitor highlights that the G2 arrest phenotype is likely triggered by the activation of this checkpoint, due to accumulating DNA damage in the absence of TRAIP caused by perturbed resolution of Replication-Transcription conflicts.

10. A similar comment regarding senescence: Why does increased polII-mediated DNA damage lead to senescence (and not, for example, apoptosis)? Can transcription inhibition suppress senescence induced by TRAIP depletion?

We have now added a section in the discussion that discusses this aspect: what is G2 senescence and why it may be appropriate that it is induced by the low level of damage created by TRAIP degradation. We attempted to rescue the G2 arrest and senescence phenotype by transcription inhibition, however, transcription itself is essential for cell cycle progression and we cannot treat TRAIP-mAID depleted cells with triptolide for too long as it blocks progression of S-phase cells into G2/M. The short triptolide treatments used do not impact this cell cycle progression, but are short enough that TRAIP-mediated DNA damage is still created in late S-phase following removal of the transcription inhibitors, leading to G2 arrest (data not shown). Further work is needed to determine what other processes / treatments can reduce the level of replication – transcription conflicts at bi-directional promoters that could be used here in more specific way. Simple resolution of R-loops by overexpression of RNase-H1 does not suppress the DNA damage phenotype (Supp Figure 11D).

Reviewer #2 (Remarks to the Author):

We are glad that the reviewer thinks that: Overall, the paper is interesting, very well organized and increases current knowledge on a key factor regulating genome integrity at multiple levels.

To address reviewer's comments:

1. The first major concern regards the use of a single cell system, the HCT116 colorectal cancer cell line, for all the functional data, raising the issue of whether the observed phenomenon is restricted to the specific cell line type. Also, the data obtained using the TRAIP depletion in cell-free Xenopus egg extracts (Figure 6) are indirect (absence of evidence is not an evidence of absence). Authors should recapitulate the key data using a different cell system.

As explained in point 4 for Reviewer 1: We created a TRAIP-mAID degron in immortalised but not transformed retinal pigment epithelium RPE1-TERT cell line. We considered other cell lines but tried to avoid HeLa cell line due to their chromosomal instability, while commonly used in DNA replication studies U2OS cells are triploid, making CRISPR modifications more tricky.

We have confirmed the main phenotypes observed upon TRAIP-mAID degradation in this cell line: decrease in cell viability (Figure 1B), cell proliferation (Supp Figure 2C), accumulation of DNA damage foci (γ H2AX and 53BP1) upon TRAIP-mAID degradation (Supp Figure 5A), accumulation of DNA damage specifically in S-phase cells (Supp Figure 8C) and dependence of this damage generation in S-phase on presence of RNA PolII on DNA (Figure 5B).

2. The second major point that needs better investigation is the mechanistic understanding of TRAIP's role in resolving replication-transcription conflicts. Is this function rescued by the re-expression of TRAIP in the AID system they have developed? Does it depend on its ubiquitin ligase activity? The use of a RING finger mutant of TRAIP will easily help address this point. If the RING finger is required, what are the targets involved in this function?

We appreciate the importance of this point. Rescue expression of TRAIP is challenging as too high level of TRAIP is also detrimental for cells and transient overexpression does not work. We have turned therefore to the retroviral system used previously by Andrew Jackson and Grant Steward's groups (Harley et al 2016). With that, we could rescue the cell viability, proliferation and G2 arrest (Figure 1 C, E and G) of TRAIP-mAID cells treated with auxin with expression of full length wild-type TRAIP but not the E3 ubiquitin ligase dead mutants used in their study. All these suggest that the mechanisms studied here depend on E3 ubiquitin ligase activity of TRAIP.

Looking for targets for TRAIP E3 ubiquitin ligase activity, we investigated possibility of RNA PolIII as a target. RNA PolIII removal from chromatin upon DNA damage is driven through its ubiquitylation and unloading by p97 segregase and a number of ubiquitin ligases has been shown to ubiquitylate RNA PolIII, as we now explain in the text. We are now showing that recombinant TRAIP can ubiquitylate RNA PolIII *in vitro* (Figure 7B), proximity of TRAIP and RNA Pol II in S-phase nuclei *in vivo* (Figure 7A), while the level of ubiquitylated RNA PolIII on chromatin during S-phase is reduced upon TRAIP-mAID degradation (Figure 7D) suggesting that TRAIP is likely to ubiquitylate RNA PolIII.

3. Figure 2D: in the paper authors show that damage occurs in the S phase upon TRAIP depletion. Since 24h after IAA induction cells already accumulate in G2 (as shown in Figure 1D), it is likely that the resulting foci count (for γ H2AX and 53BP1) refers to that of cells arrested in G2. To clarify this point, the authors should do the same IF analysis upon EdU labelling (since a similar approach has already been used in Supplementary figure 6A) and/or G2 specific markers.

We have shown in Supp Figure 8B that upon synchronous cell progression through S-phase without TRAIP (TRAIP degraded at the end of G1 arrest and cells released into S-phase) we can observe largest proportion of cells with γ H2AX foci at 16 h post release (middle/late S-phase as per DNA content analysis). We used this experiment to decide when to perform γ H2AX ChIP-seq experiment. We have now also conducted analysis suggested by the reviewer – non-targeted control in Figure 7D shows level of DNA damage repair foci in cells positive for EdU upon TRAIP degradation.

4. Figure 3A: authors must include earlier time points (and also additional information stating the proper S phase staging, e.g. PCNA profile) in order to show whether the S phase progression is unaffected upon TRAIP depletion. The observed damage accumulation observed in Figure 2D might be mainly due to the faster replication rate.

We have now included earlier timepoints to analyse S-phase entry and progression - we degraded TRAIP at the end of G1 arrest, released cells from lovastatin arrest and monitored their cell cycle progression into S-phase at earlier time points than previously. These data are now presented in Supp Figure 6A and show no differences between -/+ IAA samples. We also observe no significant changes in BrdU incorporation level during S-phase and MCM7 chromatin unloading patterns (Supp Figure 7A and C). Previous studies, using downregulation of TRAIP with siRNA over a number of days, found general slowing and asymmetry of replication forks especially in response to replication stress (Harley et al 2016, Hoffmann et al 2016 etc). None of these observations suggest faster replication rates that could potentially explain damage accumulated in Figure 2D.

5. Figure 8: authors showed that ATR inhibition in TRAIP depleted cells causes elevated DNA damage but can rescue G2 arrest (Supplementary figure 5A). In the last sentence authors propose the combinatorial treatment of ATRi and TRAIP depletion, which appears a bit contradicting with their own findings. Please explain and discuss better this concept.

We have decided to take out the additional figure about ATM/ATR and that point of discussion.

6. Supplementary figure 5A: Authors showed that ATRi but not ATMi can rescue the G2 cell cycle arrest upon TRAIP depletion, proposing that damage originating from S phase. Authors should show the activation of the canonical ATR substrate upon TRAIP depletion (Western blot showing pRPA, pCHK1 etc...).

We find that the level of DNA damage generation in S-phase upon TRAIP degradation is relatively low as shown by modest levels of γ H2AX and 53BP1 foci observed in Supp Fig 8B, Figure 5A, Supp Figure 10B. In results the level of S-phase checkpoint activation is also modest. It is enough to block cell cycle progression and the low level of activation is the most likely the reason why these cells undergo senescence rather than apoptosis (as explained now in discussion). However, it is difficult to detect this level of S-phase checkpoint induction by western blotting (data not shown). We would like to note, however, that slightly higher levels of γ H2AX and pS345-Chk1 phosphorylation was shown by western blotting in HeLa cells treated with siRNA against TRAIP (with no additional replication stress) (Harley et al 2016) and slightly increased pS317-Chk1 was shown in U2OS cells treated with siTRAIP (Feng et al 2016). We believe therefore that our attempts to gain the beautiful western blots will be again considered repetition of already published phenotypes.

Minor comments:

1. Figure 1B: colour coding for the graph is missing.

Amended – thank you

2. Figure 1C: Y axis should be better explained. The fold change after 24 hours of +/- IAA already shows almost 5-fold change.

The cells were normalised to the seeding density. We have changed the axis description to better reflect the data.

3. Figure 1D: for DNA content, authors need to mention if specific model system is applied (example: Watson pragmatic model)?

We now explain the way the gates were applied in materials and methods and also provide an example of gating in Supp Figure 2D.

4. Figure 2A: scale bars disturb the IF images. Show only bar (better 10 μ m) and mention the dimension in the legend.

We appreciate the reviewer's comments, that the pictures would look better without the dimension, however the bar with dimension is generated in our microscopy software and not added later. We cannot therefore remove the dimension without selection of a whole set of new photos for each experiment that do not contain the bar applied. We decided therefore to retain the dimension within the photos.

5. Figure 3: to substantiate the finding that DNA damage originates from S phase upon TRAIP depletion, authors should monitor γ H2AX.

We have shown in Supp Figure 8B that upon synchronous cell progression through S-phase without TRAIP (TRAIP degraded at the end of S-phase and cells released into S-phase) we can observe largest proportion of cells with γ H2AX foci at 16h post release (middle/late S-phase as per DNA content analysis). This experiment mirrors the experiment in Figure 3B but monitors γ H2AX accumulation specifically during S-phase progression.

6. Figure 4A: present the representative image showing γ H2AX hotspot with TSS bidirectionality.

The example we had previously had a nonsense transcription detected in one of the directions. We have now selected a bi-directional hotspot with annotated genes in both directions, and presented another example in Supp Figure 9A.

7. Supplementary figure 9B: authors used RNase A treatment as control for S9.6 antibody. Since RNase A is known to degrade free RNA, authors should report the exact experimental conditions used for the assay. Authors should also include RNase H treatment as control.

Thank you for spotting this. We have repeated the experiment with correct RNase H as a control. Presented now in Supp Figure 11C.

Reviewer #3 (Remarks to the Author):

We are happy to see that the reviewer thinks that:

The authors have developed an elegant and robust method to deplete untagged or mClover-tagged TRAIIP in HCT116 cell lines.

In reply to reviewers comments:

controls such as caffeine and aphidicolin were not described in the text at all, rather the method, rationale and sometimes their conclusions were described in figure legends.

We have now added description of activity of aphidicolin and caffeine in the text on page 39. We apologise for the prior lack of description of our controls in the text. As the reviewer remarked there is a lot of data in our manuscript, some of which are validating the systems we use, and are not novel findings as such.

Titling an entire figure with a conclusion is appropriate but doing so for each panel is not (Figure 3). We have removed conclusions from the first line of each panel of each figure legend description.

Further, the lack of agreement between published literature, *Xenopus laevis* extracts and HCT116 cells for mitotic experiments was only briefly stated and then not discussed in either the results or discussion. Together, this made it difficult to interpret these experiments.

We have now added more discussion of mitotic roles for TRAIIP in discussion section. We do not believe that there is a lack of agreement between published literature, extracts and our observations. *Xenopus* egg extract does not support transcription during S-phase and thus TRAIIP is not needed for completion of unperturbed DNA replication in this system. If S-phase is fully completed with no retained replication forks or replisomes on chromatin, TRAIIP is likely to be dispensable for mitosis in egg extract. However, if any replisomes are retained then TRAIIP is needed to remove them.

The situation in human cells seems to be similar – TRAIIP has been shown to be important for resolution of unfinished replication in mitosis (essential for MiDAS, Sonnevile et al 2019). It is reported that it regulates mitotic progression too although it is not clear whether this relates to its replisome removal role or not. The main difference is that TRAIIP is essential for unperturbed S-phase, which is, we believe, due to the fact that replication has to coincide with transcription, which leads to conflicts.

Other issues such as not explicitly defining abbreviations in the text or even in the figure legends (see points 7-Cod,Hth and 16-CULi) put the onus on the reader to make assumptions to follow the manuscript.

We have added more descriptions of these acronyms.

While the experimental system and initial figures were presented elegantly, the manuscript became disorganized later on; multiple supplementary figure panels were not referred to at all, details missing from other figure panels, and referencing figures out of order.

We have found one panel not referenced and corrected it (please note that some supplementary panels are mentioned in materials and methods as validation). We apologise for missing details and hope we found them all. We aimed to put together figures to contain related panels that may be given a common caption. Most of the time they follow the flow of the text, too, but rarely we may refer to a figure that was presented earlier or later.

Further, placing p values in long lists at the end of a figure legend (sometimes 8 p values in a list for one panel) means the reader must do the work rather than having the figure speak for itself.

P values in figure legends are the policy of the journal.

It is difficult to follow the argument of a manuscript when an entire figure with three panels (24 images with 2 drug treatments and 4 separate graphs with 6 bars each) is introduced in the second-to-last sentence of the discussion. This means many data panels are not properly presented in the results.

We have removed this figure from the manuscript.

Major comments

1) Line 64-65: "Genome-wide, such unreplicated regions correlate with common fragile sites (CFS), which are chromosomal loci responsible for the majority of rearrangements found in human disease"

- This sentence is incorrect due to citation of a review that does not accurately reflect the literature they cite. The review refers to Burrow et al 2009 for their overly broad statement "Strikingly, recent cancer genome-sequencing projects have revealed that more than half of the DNA rearrangements in cancers cluster within certain chromosomal loci known as common fragile sites (CFSs)..." which is not accurate. Rather Burrow et al 2009 uses fragile sites as a term for both common and rare – 52% of translocation breakpoints mapped to fragile sites -- thus this sentence is incorrect on 3 levels. 1) they only looked at translocations with gene pairs, 2) they compared to common + rare fragile sites and 3) this is only for human cancers not human disease. Delete or rewrite citing the primary literature and what was found.

We would like to thank the reviewer for spotting this misunderstanding. This review indeed defines CFS as translocation breakpoint etc. We have now cited more appropriate references for the definition of CFS we had in mind.

2) The pan-nuclear H3S10-phospho staining looks strange. Mitotic H3S10 phosphorylation normally starts in discrete foci and then signal "spreads" as chromosomes condense. This means the signal usually looks like worms or squiggles, or other textures associated with condensing chromosomes. The signal here looks like DAPI.

The DAPI staining also does not reflect a traditional mitotic cell with condensed or condensing DNA, and rather looks like an interphase cell. CENPF also marks G2 cells.

We have now presented an example of more "traditional" mitotic cell rather than early mitotic cell.

3) Line 534-536 – This sentence is confusing: "Whilst overall lower proportions of cells were detected incorporating EdU/BrdU, no differences in the average amount incorporated per positive cell were found (Supplementary Figure 6A,B)."

We have rewritten this sentence:

Lower number of cells was detected incorporating EdU/BrdU upon TRAIP degradation, suggesting fewer cells in S-phase. However, the cells that were in S-phase and labelled with BrdU or EdU, show the same level of the analogue incorporation, suggesting similar rate of replication progression independent of TRAIP degradation status (Supplementary Figure 7A,B).

Supplementary Figure 6A: What is being measured in the graphs?

- The Y-axis in Supplementary Figure 6A graph not clear, is it missing a word? Mean cells positive for EdU \diamond is this the % of cells (because this is FACS analysis I assume it is this), or # of cells? Or is it mean BrdU intensity? The red does appear shifted down in -/+IAA overlay but this would be

intensity which authors state is unchanged.... The graphs vs data are confusing— if S6A graph is the % of cells, show the gates on the data with #s like done for H3S10P in Supplementary figure 3A. How the graphs in Supplementary figure 6B reflect the primary data from the images is much clearer – the Y axis in top graph clearly states what is being measured.

We renamed the axes. We also provided one of the other replicates of this experiment as an example. Based on the literature and importance of TRAP for replication fork progression during replication stress, the aim here was to determine if there is a defect in replication, which we do not see. We do appreciate that the example suggesting higher incorporation can be misleading. This was however not a reproducible finding as our quantification shows.

4) Line 466: "... G2 cell cycle arrest mediated by WEE1."

◇ Change to "dependent on Wee1"? The arrest may be initiated/mediated by other components but the G2 arrest is maintained Wee1 so indeed it is required... and agree inhibition reduces G2 levels down to the -IAA control.

changed as suggested

5) Figure S3 – please show representative image of condensed chromosomes to pair with S3B S3D - is this the same antibody used in Figure 2A (mitosin) and now using cenpF? Be consistent in naming scheme. Also there is no mention of S3D in the text.

Mitosin and CENPF are the same protein and different antibodies are named by either name. As we needed antibodies raised in different host species for different combinations of staining sometimes we used antibody named Mitosin and sometimes CENPF. We have now uniformed the naming scheme and explained in Materials and methods which antibody is used when. S3D is now mentioned in the text.

6) Figure S3 – When did you add Wee1, after 24h of IAA or in the last 1/3 hours of a 24h treatment? Can you make a scheme?

It is added for the last 1/3 h. We added a scheme.

7) Line 489-91: "Moreover, inhibition of ATR, but not ATM, was able to rescue the observed G2 cell cycle arrest (Supplementary Figure 5A) suggesting that the damage is created in S-phase during DNA replication."

This is not necessarily true, as ATR is also involved in G2/M checkpoint activation & inhibition of ATR activity can also abrogate replication-independent G2/M checkpoint initiated by UV (Stiff et al 2008).

It is true that ATR can be involved in initiation of G2/M checkpoint also outside of S-phase. However, the S-phase role is coming first and later in the manuscript we show that degradation of TRAP in G2 stage of the cell cycle is not sufficient to lead to G2 cell cycle arrest. Altogether, it is much more likely that the ATR involvement in the G2 arrest does come from its role in S-phase. We have now added word "likely" to our statement (is likely created in S-phase) to soften this conclusion.

8) 547-549 – Conclusions and results description here are tangled and should be separated. It would help to state results independently, then state interpretation.

We have replaced words: "As per our previous data" with "Reassuringly" as this is what we meant it to mean.

9) Supplementary 7A – it is not explained at all why they use APH and what they are comparing.

Please describe

Description added in the text:

Similarly, we could observe no induction of phosphorylated Chk1 in the nucleoplasm in the absence of TRAP, while such a signal was readily induced by treatment of replicating extract with inhibitor of replicative polymerases (aphidicolin) (Figure 6C).

10) Include the name of the gene and the chromosomal coordinates shown in figure 4A – the data will become public therefore readers should be able to assess the loci presented with available data. We have changed the example as requested by another reviewer and presented our new examples with chromosomal coordinates.

11) Lines 588-591: Data presented in 4F and in the text is confusing and does not match. The numbers 75% for hotspots and 65% for low is not much of a difference but this is not even reflected in the graph. We must assume Cod stands for codirectional... Hth for head-to-head... Please define all acronyms in the figure legend text.

◇ Do the #s reflect Cod1 + Hth1? If so, you must state this clearly.

* There are no P values stated in the text or figure legend, is this significant? If it is not, please state. We apologise for the confusion with this section, we have now included this additional information in the text. The 75% for hotspots vs 65% of the low gH2AX refers to the sum of all first genes from the origin, combining together codirectional and head-to-head ones. Moreover, we have calculated the significance of this enrichment and the p-value is mentioned in the text.

12) When the authors talk about antisense transcription are they talking about actual antisense transcription where the opposite strand of the annotated gene is being partly transcribed or is it bidirectional transcription like they show in Fig 4A? Can they show browser tracks that show the antisense portion? Many promoters are actually bidirectional, with genes on both the - and + strands; these regions may be enriched there.

It is not clear if the signal from PROseq is antisense transcription in Fig 4A. It may be an unannotated (possibly abortive) transcript going the opposite direction of the annotated gene. Without clear information on the scale of the regions shown (it seems the tick marks are 1 kb apart but the font is so tiny), it is difficult to gauge.

We are now presenting more examples of genes with increased γ H2AX peaks at the TSS with indicated transcripts in opposite direction, and have clearly labelled size of the region shown and genomic coordinates. In general terms, we often find an annotated transcript proceeding in the opposite direction, so these would be bidirectional promoters, but in other cases there is no known annotated transcript, so we would consider it to be transcription start site-associated antisense transcription (Core et al., Science 2008).

13) Maintain consistency between figure panels:

- Figure 5A % of cells positive for 53bp1 vs. 5B % of cells with 3 > foci – please sure same measurements for both. If they are the same, please use the same Y axes for both

Both these figures now have no of foci per cell plotted and Fig 5A Y axis is now changed to: Percentage of cells >0 foci.

14) Figure 5B – the graphs for 53BP1 do not match shown pictures -

– this may be problem of intensity but this should be described – something is off.

This is now Supplementary Figure 10B. We have replaced the example pictures with more equal intensity.

15) I do not think you can say DRB and TRAP act in same pathway because DNA damage is not additive since the DNA damage is being induced in S phase. The #s for DRB alone are already 50% of cells... It that the majority of cells without DNA damage are in G1?

In this experiment we have arrested cells in G1 and released them into S-phase. DRB was added 12 h later, upon S-phase entry. The majority of the cells in the population being analysed are in S-phase, so the cells without DNA damage should not be in G1. We have, however, removed our comment from the text.

16) Figure 5C – pictures need to be scaled the same with scale bars in same place.

- need to count foci only in nucleus (DAPI-overlapping), not just cell... there are dots outside nucleus as well. Was this done? If so, please state.

The pictures are fuzzy and not in focus. Show higher resolution images?

Only foci within nucleus were counted. This is now explained in materials and methods. We think, that the lower resolution of the image comes from the fact that, to do PLA assays with CSK-extracted HCT116 cells, the extracted nuclei were stained in solution and then spun onto slides. In result there is more variability of planes between cells and the resolution is less crisp compared to when PLA assays are carried out on cells adhered to glass cover slips.

17) Line 698 (and throughout) - While the authors suggest TRAIIP's role in S phase is the primary role for viability, this is not strictly shown as they did not actually perform viability experiments with cells depleted for TRAIIP only in G2/M. They do show that its role in S phase is responsible for triggering the G2 arrest, therefore this referee agrees that the S phase role is very likely why TRAIIP is essential. Thus, author conclusions should be softened from "demonstrate" or "we find" to "strongly indicate" or "we conclude".

Changed as suggested.

18) Line 707-708 – replication-transcription collisions were not measured – sites of co-occurring replication and transcription were identified – potential interactions were increased.

We have now measured additionally proximity of RNA Pol II and And-1 and RNA Pol II and PCNA, but we do appreciate that this is still measuring proximity of replication and transcription machineries rather than collisions per se. We changed the text to indicate this.

19) Line 639-641 / discussion: TRAIIP may not be involved in the resolution of conflicts – this has not been demonstrated. Another possibility is that it prevents transcription-replication collisions from occurring in the first place. Prevention would lead to a similar outcome as promoting repair. This (and other?) alternate possibility should be discussed. Either way it is interesting.

Thank you. We have now directly specified these two possibilities in the text. Moreover, we have now added many more experiments addressing the mechanism of TRAIIP's action. As we do not observe change in neither global replication nor transcription, nor increase in R-loops formation, we believe that it is more likely that TRAIIP helps resolving collisions rather than preventing them.

Hopefully with the new data it is now clearer.

20) Refer to the figures in order and describe all results in the results section — supplementary figure 9b is in the discussion but never mentioned in the text.

This figure is now removed

21) Supplementary figure 9 uses inappropriate control. RNase A digests ssRNA and to a lesser extent RNA:DNA hybrids – not sure why or how this is used as a control. The S9.6 antibody when used for IF can recognize other RNAs as well as RNA:DNA hybrids (see Smolka et al 2021). More appropriate controls would be RNase H to specifically digest hybrids and Rnase T1-resistant signal? A dot blot, slot blot or gel electrophoresis a better way to measure bulk RNA:DNA hybrids.

We have now provided the correct control for S9.6 antibody experiment (Supp Figure 11C). We appreciate that despite being used as a go to standard in the field the S9.6 antibody has its problems. We have also now overexpressed GFP-RNase HI and detected no change with the damage

signal created by TRAP degradation (Supp Figure 11D). As both these experiments suggest no change/dependence on RNA:DNA hybrids we stopped at these two experiments.

22) The S9.6 staining looks different from other published versions – including HCT116 cells. Normally S9.6 clearly lights up the nucleoli – here the staining is faint and throughout nucleus (as well as in cytoplasm, which is normal).

We repeated the whole experiment from fresh with better controls etc. The result is the same. It is possible that as we use cells synchronised in S-phase rather than asynchronous cell population the overall staining pattern is a bit different.

23) Figure 8 is after the model and only mentioned in the last sentence of the discussion! What is happening with the organization?

Removed now

Minor comments

1) Figure 1b – needs legend
corrected

2) Need to cite Addgene plasmids in the manner dictated by the website – “kind gift from ...” and often the paper used.

The creator of the Addgene plasmid is a co-author of this manuscript, so it did not need to be gifted. We cited the Addgene number so that future users can find it easily.

3) Supplementary 5C: move p-value lines over the appropriate bars in graph, they seem to have gotten shifted out of place. Check right pane

Corrected. P values in the legend are requested by the journal

4) Line 704-706 – a lot of commas. The complex sentence structure makes it hard to follow the line of thought.

It is now split into 2 sentences.

5) Continuous repetition of “auxin (IAA)”. I recommend altering the first instance to auxin (+IAA). Once +IAA is defined, in it no necessary to put auxin every single time in parentheses.

We removed (IAA) throughout the text. Kept it only in legends and materials and methods.

6) Define abbreviations – CULi – link to description in the text

This supplementary figure 12A is now cited in the text. CULi is explained in Figure legend.

7) Figure 6C – add explanation of APH and APH + caff in main text as positive control? This was a helpful control but only described in legend. Label blot properly for Chk1 as phospho-chk1.

Added in the main text as: Similarly, we could observe no induction of phosphorylated Chk1 in the nucleoplasm in the absence of TRAP, while such a signal was readily induced by treatment of replicating extract with inhibitor of replicative polymerases (aphidicolin) and inhibited by ATM/ATR inhibitor (caffeine) (Figure 6C).

8) Figure 2B, Figure 3, and others: Having over 5 p values listed at the end of figure descriptions in legend makes it slow to match each p value to each condition. Either place the precise p value over the bar it represents instead of the asterisks (especially since the asterisks are not defined in the figure legend) OR define the p value asterisks as > 0.05 or >0.005, etc.... Either one would make it easier to decipher.

P values in the figure legend are requested in that format by the journal.

9) Mention that 2 independent clones were examined throughout in main text at least once? Only stated in figure legends.

Added at the beginning of the results section: Two clones of each HCT116 TRAIP-mAID and TRAIP-mAC were generated and used throughout this work to ensure that the observed phenotypes are not CRISPR off target effects.

Villa, F., R. Fujisawa, J. Ainsworth, K. Nishimura, A. L. M. Lie, G. Lacaud and K. P. Labib (2021). "CUL2(LRR1) , TRAIP and p97 control CMG helicase disassembly in the mammalian cell cycle." EMBO Rep **22**(3): e52164.

REVIEWER COMMENTS

Reviewer #1 (Remarks to the Author):

The revised version is a much-improved manuscript. In particular, the authors present new experiments validating the proposed model that TRAIp suppresses replication-transcriptions collisions, and present mechanistic data showing that TRAIp ubiquitinates RNAPolIII. They also satisfactorily address the other comments I had on the original manuscript (including gH2AX CHIP, adding additional cell lines, cell cycle analyses etc). The authors should be commended for their efforts to experimentally address all reviewer comments. The manuscript is now solid and worthy of publication in Nature Communications.

Reviewer #2 (Remarks to the Author):

The authors addressed most of the comments raised by this reviewer. However, one key aspect that remains critical and unconvincing regards point n. 2, concerning the possible targets of TRAIp activity.

Figure 7A.

PLA images are of poor quality. PLA signal shows a high background, and many PLA foci are still detectable in +IAA. The proper controls (e.g., PLA performed using only 1 primary antibody) should be included.

Figure 7B.

But the most problematic part concerns the experiments on RNAPolIII ubiquitination (Figure B and C). In Figure 7B, the quality of RNAPolIII is poor, and the appearance of a smear signal, detected by the RNAPolIII antibody, is not sufficient to suggest that it is RNAPolIII ubiquitinated by TRAIp, in the absence of further controls and experiments. For example, a higher molecular weight smear also appears in the sample without TRAIp, although the overall signal is lower because less RNAPolIII is present in the reaction, according to the RNAPolIII blot.

The detection of ubiquitinated RNAPolIII should be done by performing ubiquitin

immunoblot on purified RNAPolIII after the in vitro ubiquitination assay. Furthermore, the use of the ubiquitin mutant unable to generate chains (with the 7 lysines mutated to arginines) should be included, as also used in Wu et al, 2019 (doi: 10.1038/s41586-019-1002-0).

Moreover, it is not clear whether RNAPolIII is FLAG-tagged or 6xHis-tagged. The confusion arises from the fact that in the methods (p. 20) and in the text (p. 41) it is indicated as HIS-tagged, whereas in Suppl. Fig. 13B FLAG-RNAPolIII immunopurified with M2 FLAG beads is indicated.

Also, it is not mentioned why SUMO-TRAIP is used in the experiment. Better explain the reason and add a reference.

Figure 7C.

The representative image shows no reduction in the chromatin-associated ubiquitinated p-RNAPolIII signal and is not consistent with quantification.

As control, the RNAPolIII and pRNAPolIII immunoblot of the input (before the GST-MultiDSK pulldown) should be added to display the size of RNAPolIII. Since the GST-MultiDSK pulldown should only bind ubiquitinated proteins, the fraction of RNAPolIII present in the pulldown should be mainly ubiquitinated, resulting in an increased molecular weight detectable by Western blot comparing the input with the pulldown samples.

Overall, I strongly suggest removing all data on the ubiquitination of RNAPolIII, as further investigation is needed to strengthen the result. Otherwise, there is a risk of giving misleading information to the reader. Modify the model accordingly, adding back the question mark over the effect of TRAIP on RNAPolIII.

Reviewer #3 (Remarks to the Author):

Overall, the authors have greatly improved the manuscript and addressed most of our concerns. The manuscript reorganized some key figures and deleted others greatly improving the flow of the manuscript. They have also added data to support their conclusions. There was one key point that was not addressed. We have placed an asterisk

by these points. However there are a number of concerns (major points) that need to be addressed, as well as multiple typographical errors (minor points) in the text.

Major points

*1) Original major point #1 -- “...(CFS), which are chromosomal loci responsible for the majority of rearrangements found in human disease^{2,3}”

Change to something like: “...responsible for recurrent rearrangements often found in human disease.”

Again this statement is not true, and it was not amended. It is word-for-word the same as the original manuscript. CFS do not represent the majority of rearrangements in human disease, nor do the citations provided show the underlined portion. Damage at CFSs (often deletions) is over-represented in cancers but they are NOT the majority. It is still an overstatement to the point of being wrong. Adding an additional citation that does not show this is also not appropriate; they assessed at focal duplications and deletions over the areas of interest: “From this dataset, we retained only the simple genomic deletions and amplifications, for which both breakpoint termini mapped within 1 Mb of the center of a MiDAS region. This allowed us to examine the focal genomic rearrangements that mapped within MiDAS regions.”

Indeed, reference #3 uses a more accurate description and cites a paper that reports the phenomenon: “These sites are of broad interest, because they very frequently correspond to sites of genomic rearrangements in human cancers; indeed, six of the ten most common loci for recurrent focal deletions in human cancers lie within CFSs.¹²”

2) I agree with Referee 1 and 2 that the more appropriate way to describe DNA damage foci is by the graphs in the supplementary data rather than the method chosen by the authors. The method shown by the authors I recommend swapping the locations of figure 2D and supp figure 5a.

3) p30: “Moreover, inhibition of ATR, but not ATM, was able to rescue the observed G2 cell cycle arrest (Supplementary Figure 5B) suggesting that the damage is likely created in S-phase during DNA replication.” – ATRi removes G2/M block too, not only S.

p31: “Due to ATR-dependent checkpoint activation in TRAIP depleted cells and the known roles of TRAIP in S-phase, we investigated whether TRAIP was essential during S-phase.” – the same problem: ATR is essential not only for intra-S checkpoint.

*4) The authors did not rename the axes as stated, but their changes to the text for the description and new interpretation make the graphs easily interpretable so changing them is no longer necessary.

5) Supplemental Figure 8A – There appears to be missing data. 8 conditions are listed: - IAA - APH, -IAA+APH, +IAA-APH, +IAA+APH for the G1 arrest (blue arrow), and also for the G2 arrest (red) ◊ there are only 2 -IAA samples shown, and they do not have a label for either G1 or G2 ◊ thus one set of control samples are missing.

“When cells progressed through S phase without TRAIP, our results suggest that they struggled to complete DNA synthesis on time as EdU incorporation could be detected in late G2/early mitosis (Supplementary Figure 8A).”

6) “...the ChIP-Seq detected an overall increase to the γ -H2AX signal detected following auxin treatment in both repeats (example in Figure 4A and Supplementary Figure 9A).”

“...the ChIP-Seq detected an overall increase to the γ -H2AX signal at a subset of genomic loci detected following auxin treatment in both repeats (example in Figure 4A and Supplementary Figure 9A).”

A single snapshot of the browser does not show “overall increase”; it shows an increase at a specific genomic locus. Please change language accordingly or refer to another figure panel (Figure 4D or 4E?). An overall increase is not expected with such a modest increase in foci

#s, counting cells with 3 or more foci is modest for an overall increase. There is also a typo, “detected” is repeated.

7) “it is the TSS of the first transcribed gene with high levels of antisense to sense transcription that DNA damage will reproducibly arise at.”

◇ this conclusion is too strong and has not been demonstrated. ChIP-seq is an average of millions of cells, the first transcribed gene is potentially the most vulnerable/likely spot for a problem to arise.

8) “TRAIP is likely required to resolve DNA replication – transcription conflicts on chromatin” – the title should describe the results not the interpretation.

Suggestion: “Degradation of RNA polymerase II reduces DNA damage at TSSs in TRAIP depleted cells”

- still strong, and represents presented data.

9) Figure 5: p value for key comparison missing on graph and in figure legend: +IAA -TRP to +IAA+TRP (3rd blue dots to 4th blue dots, & 5th to 6th for second clone); lack of significant difference between -IAA and +IAA is not the same as showing significant difference between -triptolide and + triptolide. This is the necessary comparison to conclude triptolide “rescues” or reduces DNA damage.

10) The graph in 5C does not match other graphs in manuscript – it looks weird. Please format to match, use thinner lines.

-There are missing p values comparing +IAA to +IAA +TRP for loci other than WDR. Is Rac3 or BAIAP significant? BAIAP is not likely to be significantly different due to the high variation in the +IAA sample -- the error bars are overlapping with the +IAA+TRP sample if they

represent the SEM as stated in figure legend. These are important as the authors state “our selected TSS sites showed an increase of γ -H2AX signal after TRAIP degradation, which was rescued by triptolide treatment in three cases (Figure 5C).” I am also unclear what kind of test they are using to compare the 4 samples, as a pairwise t.test is not appropriate to compare a double treatment to two single treatments to an untreated – it is more appropriate to use Anova. Pairwise t.test underestimates the error.

10) “...analogously to the removal...” \diamond analogous to the removal

11) p42: “Conversely, siUBXD7 did lead to an accumulation of DNA damage repair foci in S-phase to a similar level as after TRAIP-mAID degradation and combining these two treatments did not further increase the level of detected DNA damage response(γ -H2AX and 53BP1 foci), suggesting that these two factors are epistatic and act in the same pathway (Figure 7D)” – there is no combination of siSPRTN and siUBXD7 treatments in Fig7D.

12) model (last 8 lines of results) should be presented in discussion, it is not results.

Minor comments:

1) p21: The beads were then resuspended in 18 μ l JS buffer and 6 μ l 4x LB, boiled for 5 min, vortexed, boiled again for 2 min and 20 μ l of the eluate analysed by western blotting using the antibody: Mouse anti-Rpb1 CTD (4H8) (Cell Signalling Technology 2629S; 1:5000 – parenthesis is absent

2) p23: EcoR1 (R6011, Promega) was purchased at stock 12 U/ μ l and added to the extract at 0.05 U/ μ l, Aphidicolin (A0781, Sigma) was dissolved in DMSO at 8 mM and added to the extract along with demembranated sperm nuclei at 40 μ M. – is that correct? That seems high.

3) Fig1C: all curves except of green and black look very similar – maybe it will be better to

change colors for more contrast?

4) Fig1G: no error bars for EV alone \diamond normalized to this

5) p28: SuppFig2C: is “relative cell growth” the same as “relative colony forming units” – if yes, than better to unify it, if not – the direct comparison is incorrect

6) p29: “The same was confirmed by mitotic indexing experiments (Supplementary Figure 3B)” – maybe it will be good idea to add “no WEE1i” like you did five rows later

7) Fig2D: delete word “bin” it is confusing where it is; the legend is sufficient without it

8) SuppFig7A: no error bars for CMV-OsTIR1 no auxin \diamond if results are normalized to this group, then axis and legend needs to state “normalized” or relative (as used in supplementary figure 2C)

9) Fig5B: instead of EdU positive cells better to show merge for DAPI, 53BP1, and γ H2AX

8) SuppFig12A: no data at all about CULi concentration and treatment time

9) Fig6C: it would be great to see the level of Chk1 total too. Also, there no time of APH and APH+caffeine treatment is shown. Also, what the level of pCHK1 and PCNA when you use caffeine alone?

10) p40: Previous research has indicated that formation of RNA:DNA hybrids (R-loops) is often the underlying... - RNA:DNA hybrids (R-loops) were mentioned already in p38

11) p42: p97 segragase – p97 segrEgase

12) p45: “Some cells undergoing DNA damage induced G2 senescence have been

shown with time to be able to escape the G2/M checkpoint, progress through mitosis and arrest in the subsequent G1 phase by the more efficient G1/S checkpoint.” – Is G1/S really more efficient checkpoint? Need a reference to make that kind of comment about checkpoint “efficiency”.

Response to reviewers

Reviewer #1 was fully satisfied with our changes and improvements.

Reviewer #2 was happy with most of our work.

They suggested that our work aiming at presenting RNA Pol II as a TRAIP substrate was not fully conclusive and suggested:

“Overall, I strongly suggest removing all data on the ubiquitination of RNApolIII, as further investigation is needed to strengthen the result. Otherwise, there is a risk of giving misleading information to the reader. Modify the model accordingly, adding back the question mark over the effect of TRAIP on RNApolIII.”

We have removed data on ubiquitylation of RNA Pol II (Fig 7B and 7C) as suggested, discussed the possible substrates within the discussion and modified the model in Figure 8.

Figure 7A.

PLA images are of poor quality. PLA signal shows a high background, and many PLA foci are still detectable in +IAA. The proper controls (e.g., PLA performed using only 1 primary antibody) should be included.

We repeated this set of experiments and we are presenting new images with requested controls, which are also quantified accordingly.

Reviewer #3

Reviewer 3 was happy with most of our changes, but also suggested a number of text changes:

Major points

*1) Original major point #1 -- “...(CFS), which are chromosomal loci responsible for the majority of rearrangements found in human disease^{2,3}”

Change to something like: “...responsible for recurrent rearrangements often found in human disease.”

Changed to “which are responsible for recurrent rearrangements often found in human disease.” as requested.

2) I agree with Referee 1 and 2 that the more appropriate way to describe DNA damage foci is by the graphs in the supplementary data rather than the method chosen by the authors. The method shown by the authors I recommend swapping the locations of figure 2D and supp figure 5a.

We are presenting now beeswax plots for most foci experiments. We would prefer not to change these two figures as Figure 2D shows HCT116 cell line, in which we conducted most of the experiments, while Supp Figure 5a is conducted in RPE1 cells.

3) p30: “Moreover, inhibition of ATR, but not ATM, was able to rescue the observed G2 cell cycle arrest (Supplementary Figure 5B) suggesting that the damage is likely created in S-phase during DNA replication.” – ATRi removes G2/M block too, not only S.

Changed to “during S or G2 stages of the cell cycle” on page 28.

p31: “Due to ATR-dependent checkpoint activation in TRAIP depleted cells and the known roles of TRAIP in S-phase, we investigated whether TRAIP was essential during S-phase.” – the same problem: ATR is essential not only for intra-S checkpoint.

Changed to: “Because cells depleted of TRAIP accumulate in G2 stage of the cell cycle due to ATR checkpoint activation, and TRAIP is also known also to play a role in S-phase, we started our investigations assessing whether TRAIP was essential during S-phase..”

*4) The authors did not rename the axes as stated, but their changes to the text for the description and new interpretation make the graphs easily interpretable so changing them is no longer necessary.

No change needed.

5) Supplemental Figure 8A – There appears to be missing data. 8 conditions are listed: - IAA -APH, - IAA+APH, +IAA-APH, +IAA+APH for the G1 arrest (blue arrow), and also for the G2 arrest (red) \diamond there are only 2 -IAA samples shown, and they do not have a label for either G1 or G2 \diamond thus one set of control samples are missing.

“When cells progressed through S phase without TRAIP, our results suggest that they struggled to complete DNA synthesis on time as EdU incorporation could be detected in late G2/early mitosis (Supplementary Figure 8A).”

We modified the schematic of the experiment to better reflect our data.

6) “...the ChIP-Seq detected an overall increase to the γ -H2AX signal detected following auxin treatment in both repeats (example in Figure 4A and Supplementary Figure 9A).”

“...the ChIP-Seq detected an overall increase to the γ -H2AX signal at a subset of genomic loci detected following auxin treatment in both repeats (example in Figure 4A and Supplementary Figure 9A).”

Changed as requested to “at a subset of genomic loci”.

7) “it is the TSS of the first transcribed gene with high levels of antisense to sense transcription that DNA damage will reproducibly arise at.”

\diamond this conclusion is too strong and has not been demonstrated. ChIP-seq is an average of millions of cells, the first transcribed gene is potentially the most vulnerable/likely spot for a problem to arise.

Changed to: it is the TSS of the first transcribed gene with high levels of antisense to sense transcription that DNA damage most often arises at and can be detected at a population level.”

8) “TRAIP is likely required to resolve DNA replication – transcription conflicts on chromatin” – the title should describe the results not the interpretation.

Suggestion: “Degradation of RNA polymerase II reduces DNA damage at TSSs in TRAIP depleted cells”

Due to change of the presented data and elimination of the experiments showing RNA Pol II ubiquitylation we did change this subtitle to: “Upon TRAIP degradation cells experience more DNA damage-generating encounters between replication and transcription”

9) Figure 5: p value for key comparison missing on graph and in figure legend: +IAA -TRP to +IAA+TRP

(3rd blue dots to 4th blue dots, & 5th to 6th for second clone); lack of significant difference between -IAA and +IAA is not the same as showing significant difference between -triptolide and + triptolide. This is the necessary comparison to conclude triptolide “rescues” or reduces DNA damage. Thank you for spotting this. We have now added missing values.

10) The graph in 5C does not match other graphs in manuscript – it looks weird. Please format to match, use thinner lines.

We have modified the graph as requested.

-There are missing p values comparing +IAA to +IAA +TRP for loci other than WDR. Is Rac3 or BAIAP significant? BAIAP is not likely to be significantly different due to the high variation in the +IAA sample -- the error bars are overlapping with the +IAA+TRP sample if they represent the SEM as stated in figure legend. These are important as the authors state “our selected TSS sites showed an increase of γ -H2AX signal after TRAIP degradation, which was rescued by triptolide treatment in three cases (Figure 5C).” I am also unclear what kind of t test they are using to compare the 4 samples, as a pairwise t.test is not appropriate to compare a double treatment to two single treatments to an untreated – it is more appropriate to use Anova. Pairwise t.test underestimates the error.

An additional repeat has been added to this experiment and results have been analysed using an Anova test as recommended by the reviewer. We can detect significant increased level of γ -H2AX in 3 of the 4 analysed genes in the +IAA, significantly reduced in two cases upon treatment with +TRP. Unfortunately, these experiments present a high degree of variability in the actual levels of γ -H2AX. We are assessing DNA damage levels arising from the lack of TRAIP at collision sites between transcription and replication machineries, that are a stochastic event and likely present only in a fraction of the population of cells. This is a very different situation from measuring for example DNA damage levels at a DSB site induced in all cells at the same time.

10) “...analogously to the removal...” \diamond analogous to the removal changed

11) p42: “Conversely, siUBXD7 did lead to an accumulation of DNA damage repair foci in S-phase to a similar level as after TRAIP-mAID degradation and combining these two treatments did not further increase the level of detected DNA damage response(γ -H2AX and 53BP1 foci), suggesting that these two factors are epistatic and act in the same pathway (Figure 7D)” – there is no combination of siSPRTN and siUBXD7 treatments in Fig7D. We clarified now that we mean auxin treatment and siUBXD7 together.

12) model (last 8 lines of results) should be presented in discussion, it is not results. Moved to discussion.

Minor comments:

1) p21: The beads were then resuspended in 18 μ l JS buffer and 6 μ l 4x LB, boiled for 5 min, vortexed, boiled again for 2 min and 20 μ l of 21st eluate analysed by western blotting using the antibody: Mouse anti-Rpb1 CTD (4H8) (Cell Signalling Technology 2629S; 1:5000 – parenthesis is absent This method is now removed.

2) p23: EcoR1 (R6011, Promega) was purchased at stock 12 U/ μ l and added to the extract at 0.05 U/ μ l, Aphidicolin (A0781, Sigma) was dissolved in DMSO at 8 mM and added to the extract along with demembrated sperm nuclei at 40 μ M. – is that correct? That seems high.

It is correct. *Xenopus* egg extract is very concentrated and inhibitors are usually used at higher concentrations than in cell based assays.

3) Fig1C: all curves except of green and black look very similar – maybe it will be better to change colors for more contrast?

We changed the colours to more variable.

4) Fig1G: no error bars for EV alone \diamond normalized to this

There was no normalisation here. Now error bars are added.

5) p28: SuppFig2C: is “relative cell growth” the same as “relative colony forming units” – if yes, than better to unify it, if not – the direct comparison is incorrect.

No, these are not the same. The ‘relative cell growth’ in Supplementary Figure 2C refers to normalised cell counts following 72 h of IAA treatment in the hTERT-RPE1 degrons and not colony assays. This is partly comparable to the decrease in cell numbers shown in the growth curve for HCT116. The results of colony assays in RPE1 degren cells are presented in Figure 1B.

6) p29: “The same was confirmed by mitotic indexing experiments (Supplementary Figure 3B)” – maybe it will be good idea to add “no WEE1i” like you did five rows later

Added.

7) Fig2D: delete word “bin” it is confusing where it is; the legend is sufficient without it

Removed.

8) SuppFig7A: no error bars for CMV-OsTIR1 no auxin \diamond if results are normalized to this group, then axis and legend needs to state “normalized” or relative (as used in supplementary figure 2C)

Changed.

9) Fig5B: instead of EdU positive cells better to show merge for DAPI, 53BP1, and γ H2AX

All shown now.

8) SuppFig12A: no data at all about CULi concentration and treatment time

The details of concentrations and timings are in materials and methods.

9) Fig6C: it would be great to see the level of Chk1 total too. Also, there no time of APH and APH+caffeine treatment is shown. Also, what the level of pCHK1 and PCNA when you use caffeine alone?

We do not have a good antibody recognising *Xenopus* Chk1. Egg extract is an embryonic system and has very stable protein levels. With the very quick cell cycles in the embryo, protein degradation is not the preferred way to regulate proteins, and there are very few proteins that are actively synthesised from maternal mRNA in the embryo. Most proteins are accumulated at high levels in the egg and used throughout the embryo cleavage divisions. From the literature and our experience, no changes to the total protein level of Chk1 or PCNA are expected. Caffeine only treatment would block all checkpoint signalling and no P-Chk1 would be observed.

10) p40: Previous research has indicated that formation of RNA:DNA hybrids

(R-loops) is often the underlying... - RNA:DNA hybrids (R-loops) were mentioned already in p38

Changed.

11) p42: p97 segragase – p97 segrEgase
Changed.

12) p45: “Some cells undergoing DNA damage induced G2 senescence have been shown with time to be able to escape the G2/M checkpoint, progress through mitosis and arrest in the subsequent G1 phase by the more efficient G1/S checkpoint.” – Is G1/S really more efficient checkpoint? Need a reference to make that kind of comment about checkpoint “efficiency”.
Statement removed.